# Multimodal epigenetic and enhancer network remodeling shape the transcriptional landscape of human beige adipocytes
Sarah Hazell Pickering [1,4], Natalia M. Galigniana [1,2,4], Mohamed Abdelhalim[1,4], Anita L. Sørensen[1], Julia Madsen-Østerbye[1], Manuela Zucknick[3], Philippe Collas [1,2] ✉ & Nolwenn Briand [1] ✉

Epigenetic regulation is a key determinant of adipocyte fate, driving the differentiation toward white or thermogenic beige phenotypes in response to environmental cues. To dissect the mechanisms orchestrating this plasticity in human adipocytes, we conducted an integrative analysis of transcriptomic, epigenomic and enhancer connectome dynamics throughout white and beige adipogenesis. Using a machine learning approach, we show that the white transcriptional program is tightly linked to promoter-level modulation of H3K27ac and chromatin accessibility, whereas the beige-specific induction of mitochondrial genes is driven by promoter remodeling of H3K4me3, underscoring distinct epigenetic mechanisms for white or beige specification. Adipocyte beiging is accompanied by a targeted reorganization of the 3D genome, characterized by increased recruitment of short-range enhancers controlling thermogenesis genes, enriched for C/EBP transcription factor binding sites. Our findings highlight the multimodal regulation of the beige adipocyte fate, driven by the interplay of chromatin state transitions, enhancer rewiring, and transcription factor dynamics.

Adipose tissue plays a central role in whole-body energy homeostasis, supported by the ability of adipocytes to modulate their function and phenotype to adapt to metabolic and thermal challenges. White adipocytes harbor a large lipid droplet in vivo and are dedicated to the storage of excess energy in the form of triglycerides, which can be mobilized as free fatty acids to supply energy during fasting periods. In contrast, beige adipocytes display smaller lipid droplets, a high mitochondrial content, and can dissipate energy through thermogenesis, enabled by the expression of uncoupling protein 1 (UCP1) and by the activation of various "futile" cycles[1]. Beige adipocytes arise within adipose tissue via de novo differentiation and/or from trans-differentiation of white adipocytes in response to environmental cues, such as chronic cold exposure, exercise or treatment with peroxisome proliferator-activated receptor-γ (PPARγ) agonists[2]. This "beiging" process results in whole-body metabolic improvement due to elevated energy expenditure, and increased clearance of triglyceride-rich lipoproteins and glucose from the circulation, making beige adipocytes a potential therapeutic target for metabolic diseases[3].

Epigenetic regulation is a key determinant of adipocyte fate and functional plasticity[4] Histone modifications modulate chromatin accessibility, transcription factor (TF) binding, and chromatin folding to control transcriptional activity and gene expression. In mouse models, the identity shift between white and beige adipocytes is accompanied by an epigenetic 'recoding' of enhancers[5], including acetylation and methylation of histones[6,7]. Although such adaptive mechanisms also likely operate in humans, the epigenetic remodeling and changes in 3D chromatin organization conferring thermogenic potential to human adipocytes are not fully elucidated.

Here, we characterized the spatiotemporal regulation of chromatin conformation during the establishment of a beige thermogenic adipocyte phenotype. Our integrated, genome-wide, analysis of gene expression in relation to chromatin state, chromatin accessibility and 3D conformation at gene regulatory sites highlights multiple levels of regulation mediating beige adipocyte identity.

[1]Department of Molecular Medicine, Institute of Basic Medical Sciences, Faculty of Medicine, University of Oslo, Oslo, Norway. [2]Department of Immunology and Transfusion Medicine, Oslo University Hospital, Oslo, Norway. [3]Oslo Centre for Biostatistics and Epidemiology, Institute of Basic Medical Sciences, Faculty of Medicine, University of Oslo, Oslo, Norway. [4]These authors contributed equally: Sarah Hazell Pickering, Natalia M. Galigniana, Mohamed Abdelhalim.
✉e-mail: philc@medisin.uio.no; nolwenn.briand@medisin.uio.no

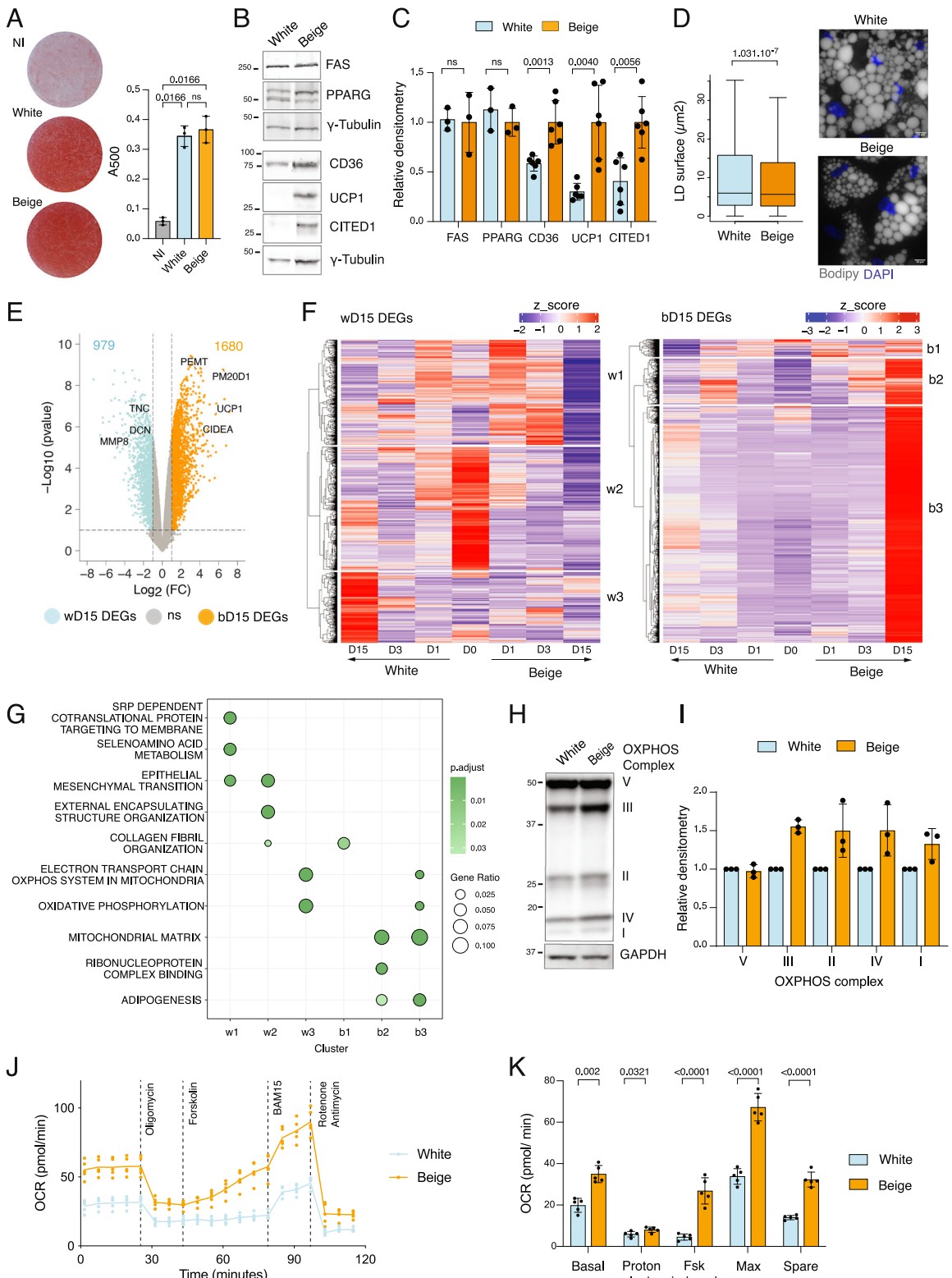

## Results

### Transcriptional kinetics of white and beige adipogenesis

Human adipose stem cells (ASCs) were differentiated into white or beige adipocytes in the absence or presence of 1 µM rosiglitazone, respectively. Both protocols lead to a similar differentiation efficiency as assessed by Oil Red O staining of neutral lipids at differentiation endpoint (day 15; D15)

(Fig. 1A), and similar protein expression of PPARG and FAS (Fig. 1B, C). However, adipocytes differentiated in the presence of rosiglitazone (hereafter beige adipocytes) show the expected increase in the protein expression of the beige markers CITED1, UCP1, and of the fatty acid transporter CD36 (Fig. 1C, D), and display significantly smaller lipid droplets (Fig. 1D). We next performed transcriptomic analysis of white *vs.* beige adipogenesis in a

**Fig. 1 | Characterization of white and beige differentiated adipocytes.**
**A** Representative Oil Red O staining (left) and staining quantification (right) of non-induced (NI) ASCs and differentiated (D15) white and beige adipocytes (mean ± SD; ns non-significant, one-way ANOVA with Holm–Šídák's multiple comparisons test; n = 3 independent differentiations). **B** Western blot analysis and **C** quantification of FAS, PPARG, CD36, UCP1 and CITED1 protein levels in differentiated (D15) white and beige adipocytes (mean fold difference ± SD; ns non-significant, two-tailed paired Student's *t* test; *n* ≥ 3 independent differentiations). γ-Tubulin is shown as a loading control. **D** Lipid droplet area analysis (Wilcoxon ranked sum test; *n* = 3 independent differentiations) (left panel) from bodipy staining in differentiated (D15) white and beige adipocytes (right panel); Scale bar: 10 μm. **E** Volcano plot of differential gene expression in differentiated (D15) white and beige adipocytes (*p* value < 0.01, |log2 fold-change | > 1; ns non-significant). **F** Hierarchical clustering of genes overexpressed in D15 white (wD15 DEGs; left panel) or beige (bD15 DEGs; right panel) adipocytes, scaled across the differentiation time-course. **G** Comparative overrepresentation analysis of wD15 and bD15 DEGs clusters from **F** using GO, Reactome and Hallmark gene sets from MSigDb. **H** Western blot analysis and **I** quantification of OXPHOS complex proteins levels in differentiated (D15) white and beige adipocytes (mean fold difference ± SD; n = 3 independent differentiations). GAPDH is shown as a loading control. **J** Oxygen consumption rates in white and beige adipocytes under basal conditions and after sequential drug treatments (mean ± SD; *n* = 5 replicates). **K** Basal, Forskolin-induced, maximal and spare respiration and proton leak quantified from (**J**) (mean ± SD; *p < 0.05, ***p < 0.0001, unpaired two-tailed Student's *t* test; *n* = 5 replicates). Fsk Forskolin, Max maximal.

triplicate differentiation time course, starting from a common pre-induction (D0) timepoint. Principal component analysis (PCA) clearly discriminates timepoints (D0, D1, D3, D15) as well as white and beige adipocytes on D15 (Supplementary Fig. 1A). Indeed, 1680 genes show increased expression in beige adipocytes (beige D15 differentially expressed genes; bD15 DEGs), including the beige markers *UCP1, CIDEA, PM20D1* and *PEMT*, while 979 are upregulated in white adipocytes (wD15 DEGs) (Fig. 1E; Supplementary Fig. 1B; Supplementary Data 1). The sustained upregulation of key beige genes following rosiglitazone removal indicates that the beige adipocytes transcriptome reflects a distinct cellular identity rather than an acute response to PPARγ agonism (Supplementary Fig. 1C). Strikingly, a large proportion of wD15 DEGs are most expressed on D0 (cluster w2) or at early differentiation timepoints (cluster w1; Fig. 1F, left panel). Conversely, bD15 DEGs are almost exclusively late induced (cluster b2, b3; Fig. 1F, right panel). Comparative functional analysis across clusters reveals a specific enrichment for extracellular matrix genes among wD15 DEGs (cluster w1, w2), while bD15 DEGs are enriched for genes pertaining to the hallmark "adipogenesis" term, consistent with increased PPARγ activity in rosiglitazone-treated cells[8] (Fig. 1G). Interestingly, genes related to distinct mitochondrial processes are overrepresented in both white and beige D15 DEGs (Fig. 1G, Supplementary Fig. 1D), indicating a transcriptional remodeling of mitochondrial function between beige and white adipocytes in our system. Supporting this view, white and beige adipocytes display distinct stoichiometries of the mitochondrial respiratory complexes I, II, III and IV, as assessed by Western blotting (Fig. 1H, I). Functionally, measurements of oxygen consumption rates confirm higher basal, maximal, uncoupled and forskolin-induced respiration in beige adipocytes (Fig. 1J, K).

To assess the physiological relevance of in vitro beige adipocytes, we compared their transcriptional profiles with human adipose tissues. The transcriptional differences between beige and white adipocytes, including enrichment for the BATLAS brown gene signature[9] and depletion of extracellular matrix genes, mirror those observed between human brown and white adipose tissues[10] (Supplementary Fig. 1E). Furthermore, RNA-seq deconvolution using a recent snRNA-seq atlas of human adipocytes[11] reveals that white and beige adipocytes map to distinct adipose sub-populations, reflecting a shift from lipid storing to thermogenic functions (Supplementary Fig. 1F).

Altogether, these results describe cell states consistent with in vivo human adipocytes, and highlight distinct transcriptional kinetics underlying the establishment of white *vs.* beige adipocyte phenotypes.

## Differential chromatin opening at promoters and enhancers in mature beige adipocytes

To determine to what extent changes in chromatin accessibility contribute to the beige adipocyte phenotype, we performed ATAC-seq during the white and beige differentiation time courses. PCA of ATAC read counts discriminates each timepoint and shows a good reproducibility between replicates (Supplementary Fig. 2A), allowing us to call differentially accessible regions (DARs). Overall, the numbers of ATAC peaks and peak coverage are similar between samples (Supplementary Data 2). However, induction of adipogenesis elicits tens of thousands of DARs (*p* < 0.01)

resulting in a global increase in accessibility by D15 in both lineages (Supplementary Fig. 2B). Beige *vs.* white DARs only emerge on D15, with beige adipogenesis enhancing accessibility in 24,637 regions and reducing accessibility in 17,577 regions (Fig. 2A). D15 DARs are enriched in intronic and distal intergenic regions, suggesting large changes in enhancer accessibility between white and beige adipocytes, with beige DARs showing additional enrichment at proximal promoters compared to white DARs (Fig. 2B). To assess the epigenetic state of white and beige DARs, we profiled 6 histone post-translational modifications by ChIP-seq: (i) H3K4me3, marking promoters of active genes, (ii) H3K27ac, marking active enhancers and promoters, (iii) H3K4me1, marking enhancers, (iv) H3K36me3, enriched in transcribed gene bodies, (v) H3K27me3, a repressive mark deposited by Polycomb repressive complex 2, and (vi) H3K9me3 marking constitutive heterochromatin. The vast majority of beige and white DARs annotate to enhancer elements, defined in a chromatin state model learned from our ChIP-seq data (Fig. 2C; Supplementary Fig. 3), and 20% of beige DAR coverage annotate to TSSs (*vs.* 7% for white DARs). Thus, beige adipogenesis results in increased chromatin accessibility at both enhancer and promoter regions.

We then asked whether D15 DARs arose from 'closing' of early-open regions, or from de novo 'opening' of previously closed or less accessible regions. We assessed the variation in ATAC signals inside D15 DARs across white and beige differentiation (Fig. 2D). In agreement with our gene expression data (see Fig. 1F), we find that white DARs mainly arise from a reopening of regions that are highly accessible at D0 but transiently close after differentiation induction (Fig. 2D, left; clusters wa1-3); only few arise from late opening regions (cluster wa4). In contrast, virtually all beige DARs arise from late (post-D3) chromatin opening events (Fig. 2D, right, clusters ba1-3). Quantification of ATAC signals within clusters and at each timepoint corroborates these observations (Supplementary Fig. 4).

Since beige DARs are enriched at promoters, we next examined the relationship between chromatin accessibility and gene expression levels. We observe a positive correlation between differential gene expression and differential ATAC-seq signal ($R^2 = 0.53$, $p < 2.2e^{-16}$) (Fig. 2E). In addition, beige DAR-associated genes are strongly induced in bD15 adipocytes, whereas white DAR-associated genes are expressed throughout adipogenesis (Fig. 2F). Similar to white DEGs (Fig.1G), white DAR-associated genes are implicated in extracellular matrix organization and cytoskeleton interactions (Fig. 2G), while beige DAR-associated genes instead relate to fatty acid metabolic processes (Fig. 2H). Altogether, our results indicate that increased accessibility of promoters and enhancers in late differentiation shape the beige adipocyte fate.

## Distinct epigenetic modules contribute to white and beige-specific transcriptional programs

Changes in chromatin accessibility at promoters can be driven by histone modifications and/or differential transcription factor binding and activity. To assess the relative contribution of epigenetic modifications and chromatin opening at promoters to the regulation of gene expression in adipocytes, we established a machine learning model using LightGBM[12], which we interpreted with the Shapley additive explanations (SHAP) method[13]

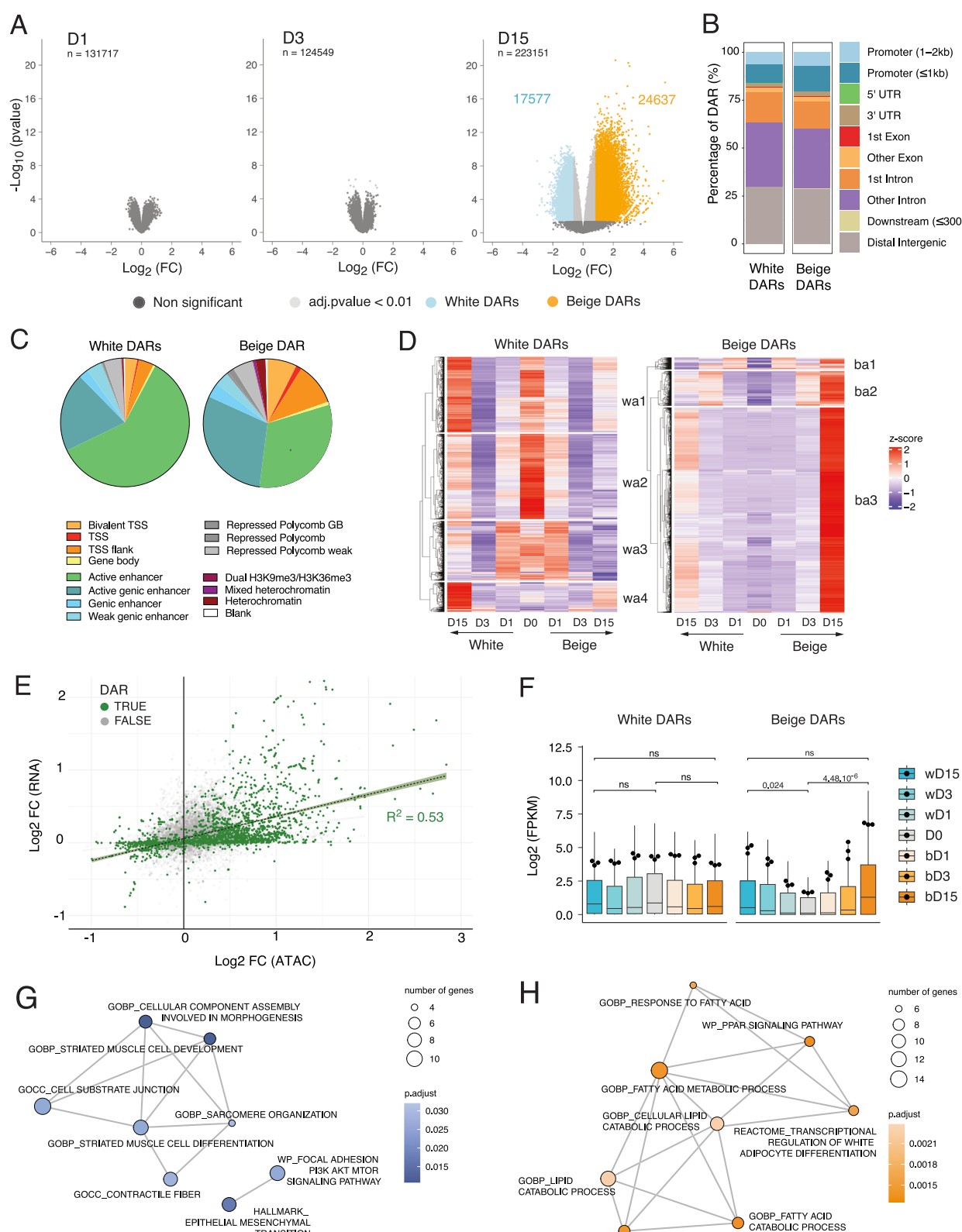

**Fig. 2 | Late opening of promoter regions in beige adipocytes. A** Volcano plot of differentially accessible regions in white (wDAR) *vs.* beige (bDAR) adipocytes at each time point; $n$ = total number of regions. The number of wDARs and bDARs identified at D15 is indicated on the plot. **B** Annotation of w- and bDARs to genomic regions. **C** Chromatin state annotation of DARs, as percentage of genome coverage. **D** Heatmap with hierarchical clustering of ATAC read counts in DARs, scaled across the differentiation time-course. **E** Scatter plot of log2 fold change ATAC signal versus log2 fold change RNA FPKM at promoters (TSS ± 2 kb) of

differentially expressed genes (D15; $p < 0.01$). Promoters with significant DARs are shown in green. **F** Expression level (log2 FPKM) of genes with differentially accessible promoters across the time-course (white DAR-associated promoters: 211 genes, beige DAR-associated promoters: 222 genes, two-way ANOVA $p = 0.0125$; $t$-test with Holm adjustment; $n = 3$ RNA-seq replicates). Boxplots represent the spread of expression between genes. Network of annotated terms enriched for genes with increased accessibility in white (**G**) or beige (**H**) differentiated adipocytes (D15).

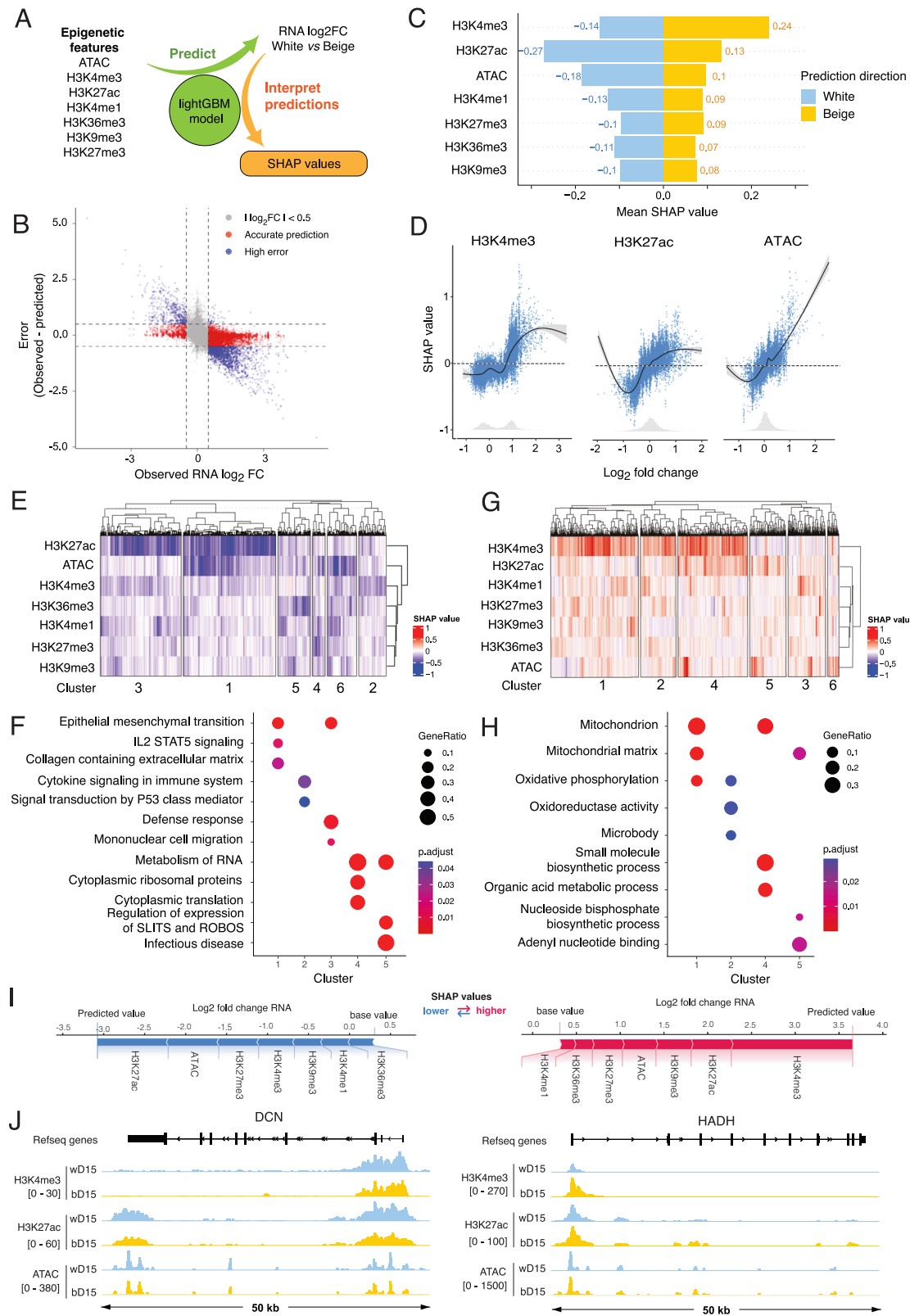

**Fig. 3 | Distinct epigenetic regulations at promoters of white and beige specific transcripts. A** Methodological flowchart for interpretation of epigenetic datasets using a LightGBM machine learning predictor explained by SHAP. **B** Scatter plot of observed RNA log2 fold change versus prediction error. Transcripts with |RNA log2 fold change | > 0.5 and |error | < 0.5 were considered for further analysis. **C** Average SHAP value for each epigenetic feature stratified by direction of the prediction. **D** SHAP dependency graphs for the predictive power of the H3K4me3, H3K27ac and ATAC features. Gray histograms show the distribution of features values. **E**, **G** Heatmap with hierarchical clustering of SHAP values for each epigenetic feature for filtered wDEG (**E**) and bDEG (**G**) transcripts. **F**, **H** Comparative overrepresentation analysis for clusters from (**E**, **G**) (**F**, white SHAP clusters; **H**, beige SHAP clusters) using GO, Reactome and Hallmark gene sets from MSigDb. **I** SHAP waterfall plots showing the effect of epigenetic features on the prediction of the expression of *DCN* (wDEG, left) and *HADH* (bDEG, right). **J** Genome browser views of H3K4me3, H3K27ac and ATAC enrichment at the promoters of white overexpressed (*DCN*: ENST00000052754.10) and beige overexpressed (*HADH*: ENST00000638621.1) genes in white and beige adipocytes.

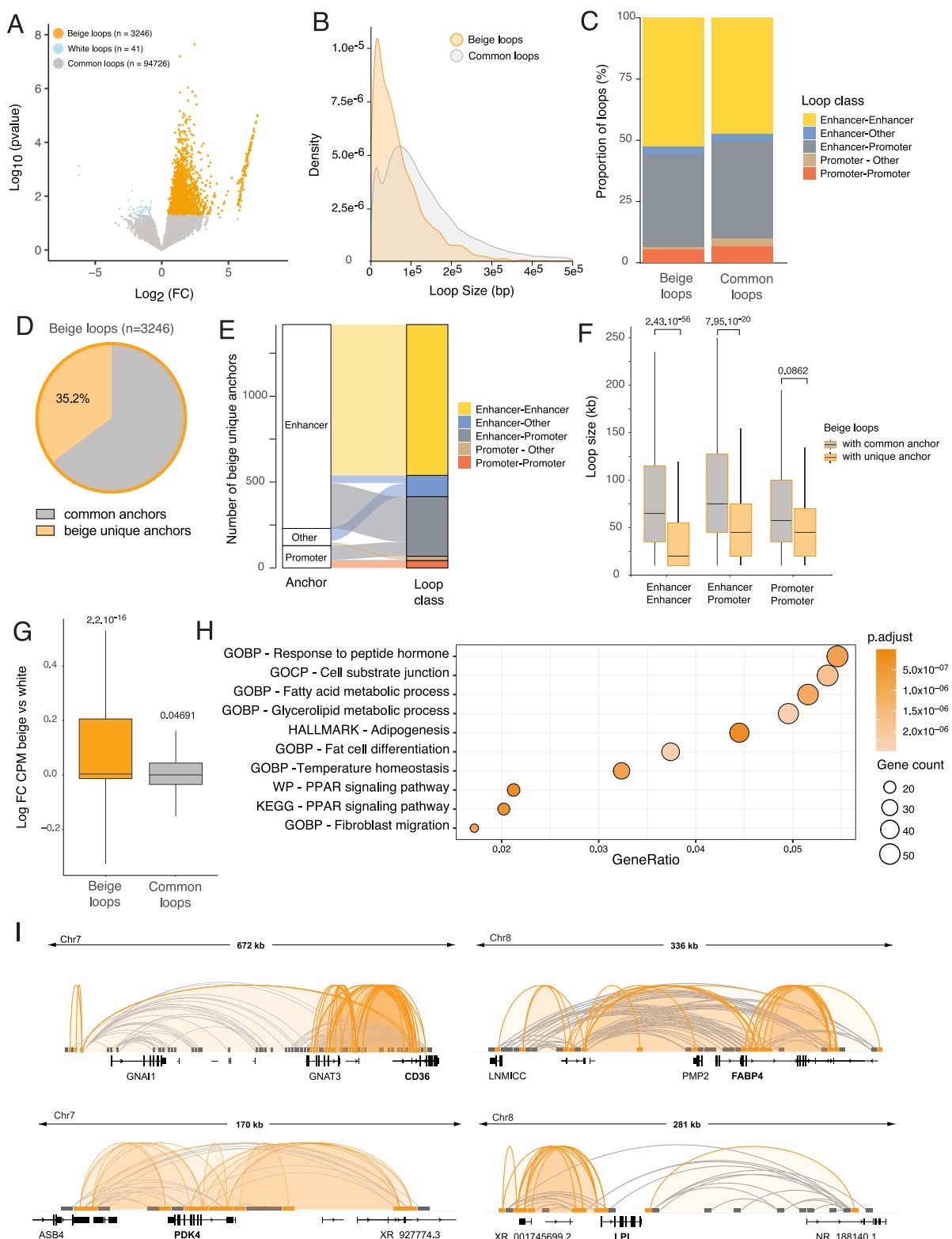

(Fig. 3A, Supplementary Fig. 5A–C). Training on epigenetic features, the model achieves a correlation $r^2$ of 0.61 on hidden data (Supplementary Fig. 5D) and accurately predicts approximately 70% of variably expressed transcripts (|log2 FC| > 0.5 and |error| < 0.5; 823 bDEGs and 182 wDEGs) (Fig. 3B). However, the model fails to predict changes in TSS expression for a subset of transcripts (1976 transcripts with |log2 FC| > 0.5 and |error| >

0.5), suggesting these are regulated by additional features not included in the model, such as enhancer regulation (Fig. 3B; see next section). We thus focused on transcripts for which epigenetic modifications at promoters successfully predict expression outcome and performed SHAP analysis to explain the contribution of each epigenetic feature to the model prediction. Strikingly, SHAP analysis highlights different modes of regulation for white

**Fig. 4 | De novo looping of short-range enhancers promote beige gene expression.**
**A** Volcano plot of differential H3K27ac Hi-ChIP signal in beige *vs.* white adipocytes (D15) ($n$ = 3246 beige loops and $n$ = 41 white loops, $p < 0.05$). **B** Density distribution of H3K27ac Hi-ChIP loop size, defined as the minimum distance between two anchors. **C** Proportion loop category for H3K27ac Hi-ChIP beige specific and common loops. Anchors located upstream of a TSS (-2kb) and overlapping with a H3K4me3 ChIP peak are defined as Promoter. Anchors overlapping with a H3K27ac ChIP peak are defined as Enhancer. All remaining anchors are defined as Other. **D** Proportion of beige loops with anchors representing de novo genomic

interactions. **E** Alluvial plot showing the proportion of Enhancer, Promoter and Other categories in de novo beige anchors and their corresponding loop annotation. **F** Size of beige loops involving common anchors or beige unique anchors for the indicated loop annotations (Wilcoxon rank-sum test) (**G**) Relative expression (log2 fold change CPM beige *vs.* white, D15) of transcripts associated with beige and common loops (Wilcoxon signed-rank test). **H** Overrepresentation analysis of genes associated with beige loops using GO, Reactome and Hallmark gene sets from MSigDb. **I** Genome browser views of H3K27ac Hi-ChIP beige (orange) and common (gray) loops at *CD36*, *FABP4*, *LPL*, and *PDK4* loci.

and beige DEGs promoters, the strongest feature for the prediction of TSS expression being H3K27ac for wDEGs and H3K4me3 for bDEGs (Fig. 3C). Indeed, SHAP partial dependence plots highlight distinct relationships between the enrichment of epigenetic features and their influence on model output (Fig. 3D): (i) increased H3K4me3 levels are predictive of higher gene expression in beige adipocytes, while this relationship is lost for wDEGs; (ii) the relationship between H3K27ac levels and SHAP values indicate a strong impact for wDEGs and a more moderate effect on model prediction for bDEGs; (iii) the full range of chromatin accessibility signal is linearly correlated with the predictive power of our model, consistent with our previous correlation result (Fig. 3D, see Fig. 2E). When assessing modes of regulation for each transcript, we confirm that H3K27ac levels are strong predictors of most wDEG expression (Fig. 3E, cluster 1 and 3), although changes in chromatin accessibility also contribute to a subset of transcript prediction (Fig. 3E, cluster 1). GO term analysis highlights pathways-specific modes of regulation, with extracellular matrix genes being controlled by both variation in H3K27ac and chromatin accessibility (Fig. 3F). For bDEGs, changes in H3K4me3 are the strongest predictor for most transcripts, in association with variations in H3K27ac levels and chromatin accessibility (Fig. 3C, G, clusters 1,2,4). GO term analysis indicates the increase in mitochondrial genes expression in beige adipocytes is closely tracked by H3K4me3 signal at promoters (Fig. 3H). Importantly, the strongest epigenetic predictors according to SHAP analysis also show the most variation at individual TSSs, as exemplified by the DCN and HADH loci (Fig. 3I, J). In summary, our machine learning model points to lineage- and pathway-specific epigenomic regulation of DEG promoters in white and beige adipocytes.

### Recruitment of short-range enhancers in beige adipocytes

Since changes in promoter epigenetic landscape alone do not explain expression variation for a subset of DEGs transcripts (see Fig. 3B), and differential chromatin opening occurs mostly at enhancers (see Fig. 2C), we next asked whether dynamic enhancer-promoter interactions also contribute to gene expression changes in white *vs.* beige adipocytes. To capture enhancer interactions at D15, we performed H3K27ac Hi-ChIP (Supplementary Fig. 6A). When calling chromatin interactions (loops) using the MAPs pipeline[14], we found a similar number of loops across conditions (counts ≥ 6; Supplementary Fig. 6B), and a complete overlap of the called loops between biological replicates (Supplementary Fig. 6C). We then called differential loops between white and beige adipocytes using HiC-DC + [15] ($p < 0.05$) (Fig. 4A; Supplementary Data 3). Amongst 98013 loops called, 3246 loops are significantly enriched in beige adipocytes ("beige loops") and only 41 loops are enriched in white adipocytes (Fig. 4A). Thus, the D15 white-specific transcriptomic profile is not mediated by distinct H3K27ac enhancer interactions.

Importantly, all beige and common loops overlap with H3K27ac peaks from our D15 ChIP-seq at one or both anchors (Supplementary Fig. 6D), validating the Hi-ChIP experiment. Beige loops represent robust interaction events, with more than 40 counts per loop on average (Supplementary Fig. 6E). Beige and common loop sizes range from 10 kb to 500 kb, typical for interaction of regulatory elements[16] (Fig. 4B). However, beige loops show a strong enrichment for shorter range interactions, which could reflect formation of proximal promoter-enhancer loops or the formation of enhancer hubs (Fig. 4B). To annotate the loops, we defined 5 classes of interactions based on the presence of H3K4me3 (Promoter) or H3K27ac

(Enhancer) peak from our ChIP-seq data within the loop anchors. Enhancer-enhancer and enhancer-promoter interactions are predominant in our dataset, as expected for a H3K27ac Hi-ChIP, with beige loops showing a slight enrichment for enhancer-enhancer interactions (Fig. 4C).

We next assessed whether beige loops arise from new interactions between anchors already involved in common loops, or from the recruitment of new genomic regions. Strikingly, 35% of beige loops are formed by de novo chromatin contacts (Fig. 4D). These beige unique anchors mainly represent enhancer regions, which predominantly connect with other enhancers and with promoters (Fig. 4E). Beige-specific enhancer-enhancer and enhancer-promoter interactions formed by de novo contacts are significantly smaller than those involving common anchors (Fig. 4F). Thus, beige adipogenesis results in both the recruitment of short-range enhancers to promoters, and in the clustering of enhancers in close genomic proximity.

Increased contacts at promoters in beige loops associate with a significant upregulation of expression of the associated genes, in contrast to genes implicated in common loops, which are not differentially expressed in white *vs.* beige adipocytes (Fig. 4G). Genes associated with beige loops are enriched for "Fatty acid metabolic processes", "PPARG signaling pathway" and "Temperature homeostasis" annotations (Fig. 4H). Indeed, we detect a strong increase in chromatin contacts at the promoters of genes involved in lipid transport (*LPL*, *CD36*, *FABP4*) and fatty acid oxidation (*PDK4*) (Fig. 4I). Consistently, these genes are strongly upregulated in beige compared to white adipocytes derived from the donor used in this study, as well as from five additional unrelated human subjects[17] (Supplementary Fig. 6F). We conclude that the establishment of the beige adipocyte phenotype is mediated by the recruitment of short-range enhancers to regulate the expression of fatty acid metabolism and thermogenic genes.

### Beige loops are enriched in early adipogenic transcription factor binding sites

De novo chromatin interactions can be promoted by changes in chromatin state and accessibility, as well as differential binding of TFs and mediator complexes[18]. We first reasoned that increased chromatin contacts in beige adipocytes detected by H3K27ac Hi-ChIP could be mediated by an increase in H3K27ac levels and/or chromatin opening at loop anchors. We find that nearly half of beige loops harbor a beige DAR at either anchor, and 25% display both increased chromatin opening and increased H3K27ac levels (Fig. 5A). However, 41% of beige loops are formed without significant variations in H3K27ac or ATAC signals at their anchors (Fig. 5A), suggesting that differential binding of protein complexes may drive beige-specific 3D genome folding at these sites[19].

The MED1 subunit has been shown to mediate ligand-dependent binding of the Mediator coactivator complex to nuclear receptors[20], promoting looping of enhancer communities to promoters in various cellular contexts, including during white and beige adipogenesis[21–24]. We thus re-analyzed previously published MED1 ChIP-seq datasets in human beige adipocytes[23] and assessed MED1 ChIP signal enrichment at beige and common loops (Fig. 5B). We find significantly higher MED1 levels at beige versus common loop anchors (Cohen's $D = 0.98$; Fig. 5B), consistent with a higher proportion of active enhancers in beige loops, driving high levels of gene expression (see Fig. 4C, G)[25].

To identify TFs potentially associated with MED1 at beige loops, we performed a motif analysis at beige MED1 peaks at beige anchors. We find a

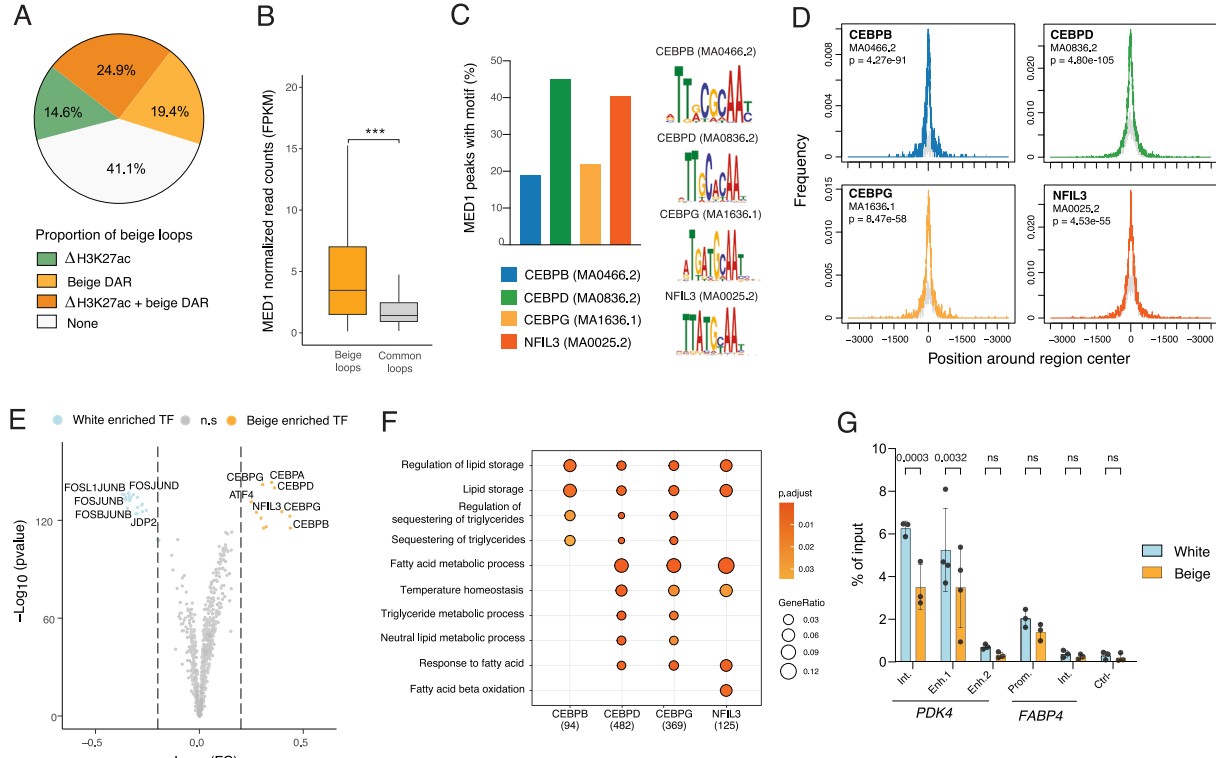

**Fig. 5 | Beige loops are enriched for early adipogenic transcription factor footprints. A** Proportion of loops with differential H3K27ac peaks (log2 beige *vs*. white fold change > 1) and/or beige DARs at either anchor. **B** Normalized read counts of MED1 ChIP signal[23] in beige loops and common loops anchors (*** | *D* | = 0.98, Cohen's D standardized mean difference). Enrichment analysis for TF motifs within MED1 peaks in beige loops anchors. For the top 4 enriched motifs, proportion of MED1 peaks with motif (**C**) and metaplot of TF motif enrichment relative to genomic regions centers (**D**). **E** Volcano plot of TF footprint enrichment in beige loop anchors in white versus beige D15 adipocytes (white enriched: log2 fold change < −0.2; beige enriched: log2 fold change > 0.2). **F** Comparative over-representation analysis of indicated TF target genes using GO, Reactome and Hallmark gene sets from MSigDb. **G** ChIP-PCR analysis of NFIL3 binding at *PDK4* intronic and distal enhancers, *FABP4* promoter and intronic region and a negative control region (two-way ANOVA with Sidák's multiple comparison test, ns non-significant; *n* ≥ 3 independent differentiation experiments).

significant enrichment for TFs from the C/EBP family, with motifs for C/EBPδ and the C/EBP-related factor NFIL3 being detected in about 40% of the peaks (Fig. 5C, D). To predict differential TF binding in white and beige adipocytes, we conducted a differential footprinting analysis on beige loops anchors using ATAC-seq from D15 white and beige adipocytes (Fig. 5E and Supplementary Data 4). Strikingly, top enriched TFs at beige loops are members of the C/EBP family, including NFIL3, and their binding partner ATF4 (Fig. 5E). In white adipocytes, the same regions are enriched for TFs from the AP-1 family (FOS, FOSL1, FOSL2, JUNB, JUND), and for JDP2, an AP-1 component that potently inhibits AP-1 mediated transcription[26]. Differential footprints for C/EBPs and NFIL3 TFs are found at enhancer-enhancer, enhancer-promoter and promoter-promoter loop anchors (Supplementary Fig. 7A). Genes associated with these differential footprints annotate to fatty acid metabolic processes and temperature homeostasis (Fig.5F), and show a global increase in expression in beige *vs*. white adipocytes (Supplementary Fig. 7B).

NFIL3 has been described as both a transcriptional repressor and activator[27,28] and can compete with C/EBPs for DNA binding[29,30]. Since footprinting analysis cannot discriminate TFs with similar motifs (see Fig. 5C), we directly assessed NFIL3 binding profile by ChIP-PCR. NFIL3 is expressed at similar levels between white and beige adipocytes (Supplementary Fig. 7C, D). In white adipocytes, we detect a strong binding of NFIL3 at *PDK4* regulatory regions, and at the promoter of *FABP4* (Fig.5G and Supplementary Fig. 8A–C). Strikingly, NFIL3 binding is significantly decreased at both intronic and distal enhancer of *PDK4* in beige adipocytes (Fig.5G). However, partial knockdown of NFIL3 in white adipocytes has no measurable impact on *PDK4* expression (Supplementary Fig. 9A–C), indicating that reduced NFIL3 levels alone are not sufficient to activate

*PDK4* transcription. Thus, decreased NFIL3 binding correlates with increased chromatin opening at those sites, increased loop formation, and transcriptional activation. Collectively, these results suggest that increased footprints at *PDK4* enhancers rather reflect an increased binding of C/EBPδ and/or C/EBPβ, and point to NFIL3 as potential negative regulator of the beige adipocyte phenotype.

## Discussion

Adipocytes are a functionally heterogeneous cell type, and the relative proportion of distinct adipocyte subtypes within adipose tissue associates with differential metabolic disease risk[31]. Beige adipocytes are particularly plastic, as these can convert between the white and beige states in response to environmental changes. Here, we show that the beige adipocyte transcriptomic signature is established late during adipogenesis, and associates with beige-specific chromatin opening at both promoters and enhancers. Distinct epigenetic modulations at promoters drives the upregulation of white- and beige-specific pathways, with beige-specific upregulation of mitochondrial genes correlating with an increase in H3K4me3 levels. In contrast, the upregulation of thermogenic genes cannot be explained by epigenetic remodeling only, but associates with the recruitment of short-range enhancers enriched for C/EBP TFs binding motifs.

The establishment of a beige-specific thermogenic program coincides with a global opening of the chromatin at the promoters of beige-induced genes. We used machine learning and Shapley values to cluster promoters based on the contribution of epigenetic features to changes in TSS expression. This approach allowed us to integrate data across modalities, investigate correlations at each promoter in an error-aware manner, and to overcome potential limitations of feature-specific analysis. We highlight

distinct epigenetic modules regulating white and beige transcriptional programs, with respectively higher predictive power for H3K27ac or H3K4me3. Intriguingly, altered levels of these epigenetic marks associate with persistently impaired adipocyte function in a model of weight loss[32]. It is thus tempting to speculate that changes in the epigenetic landscape at beige-specific promoters could similarly encode a memory of beiging, by either facilitating their reactivation or priming for enhanced transcription upon adrenergic stimulation[33]. By measuring the prediction error from our machine learning model, we inferred gaps in the model such as the influence of enhancer looping. While the promoter-enhancer looping feature could in theory be incorporated into the model, the resolution of the Hi-ChIP dataset is ill-suited for the prediction of expression at single TSSs.

Strikingly, our Hi-ChIP analysis of white *vs.* beige enhancer interactions reveals only a negligible proportion of total loops enriched in the white lineage, compared to a markedly higher enrichment of beige -specific loops. This suggests that the white adipocyte phenotype constitutes the default adipose fate for subcutaneous adipose tissue-derived ASCs, while additional stimuli are needed to acquire the beige phenotype. This is consistent with in vivo data showing that beige adipocytes revert to a white phenotype upon stimulus removal[34,35]. Our findings suggest that the increase in enhancer looping in beige adipocytes is largely independent of changes in H3K27ac levels, indicating that these regulatory regions are pre-established in white adipocytes but engage in de novo interactions through TF recruitment upon beiging stimulation. In particular, both motif and ATAC footprinting analysis point to an enrichment of TFs from the C/EBP family at beige loops anchors. This is consistent with the role of C/EBPβ as a regulator of beige adipocytes' thermogenic program[36]. In our ChIP-qPCR analysis, NFIL3 binds to enhancers of PDK4, a key metabolic enzyme that redirects glucose from oxidation towards triglyceride synthesis, favoring the use of fatty acids as energy source[37,38]. We observe that decreased NFIL3 binding at PDK4 enhancers correlates with increased transcription in beige adipocytes. These findings differ from a recent report suggesting a positive role for NFIL3 in cAMP-induced adipocyte beiging[39]. NFIL3 can act as both a transcriptional activator and repressor[27,28], depending on context, for instance through heterodimerization with other bZIP TFs such as CREB[40] or by competing with C/EBPs for binding motifs[30,41]. Therefore, the apparent differences in NFIL3's role between rosiglitazone- and cAMP-induced beiging systems might stem from the context-dependent activity of NFIL3.

It is important to acknowledge that our findings are largely correlative, and future functional experiments are needed to establish a causal relationship between genome remodeling and the establishment of a beige phenotype. In addition, given the timescale of adipose differentiation, this experimental setup lacks the temporal resolution needed to infer causality direction between (epi)genome remodeling and transcriptional changes[42]. Indeed, we cannot rule out that transcriptional activation is the cause, rather than the result, of differential loop formation. Nevertheless, our results highlight the complex epigenetic and enhancer network remodeling associated with the establishment of the beige adipocyte identity.

## Materials and methods
### Cell culture and adipose differentiation
Primary ASCs were isolated from subcutaneous fat obtained by liposuction from a female donor (age 45; BMI 20.9 kg/m[2]), obtained after donor's informed consent as approved by the Regional Committee for Research Ethics for Southern Norway with number REK 2013/2102. ASCs were cultured in DMEM/F12 (17.5 mM glucose) with 10% fetal calf serum and 20 ng/ml basic fibroblast growth factor (proliferation medium). Upon confluency, fibroblast growth factor was removed, and cells cultured for 72 h in DMEM/F12 (17.5 mM glucose) with 10% fetal calf serum (basal medium) before induction of differentiation (day 0: D0). For white adipose differentiation, ASCs were induced with a cocktail of 0.5 μM 1-methyl-3 isobutyl xanthine (Sigma, I5879), 1 μM dexamethasone (Sigma, D4902), 10 μg/ ml insulin (Sigma, I9278) and 200 μM indomethacin (Sigma, I7378-56) in basal medium. Differentiation media was renewed every 3 days until day 9, after which cells were maintained in DMEM/F12 (17.5 mM glucose) with 10%

fetal calf serum and 10 μg/ ml insulin. For beige adipose differentiation, media were supplemented with 1 μM Rosiglitazone (Sigma, R2408) until day 15. Samples were harvested on D0, then D1, D3 and D15 after induction. White and beige differentiation experiments were performed in parallel, each conducted independently in at least three biological replicates using ASCs at passage 3-8.

### Oxygen consumption
Real-time oxygen consumption rates (OCR) of white and beige adipocytes were measured in XF media (non-buffered DMEM containing 2 mM L-glutamine, 25 mM glucose, 1 mM sodium pyruvate and 2% BSA) using a Seahorse XFe 96 Analyzer (Agilent Technologies). During the assay, 2 μM Oligomycin (O4876, Sigma), 10 μM forskolin, 2.5 μM BAM15 (HY-110284, MedChemExpress), and 1 μM Rotenone/Antimycin (Agilent) were added sequentially. The quantifications presented are corrected for non-mitochondrial respiration.

### Oil Red O staining and quantification
Non-induced ASCs and D15 cells were fixed in 4% paraformaldehyde for 20 min and intracellular neutral lipids were labeled for 30 min with Oil Red-O (O0625; Sigma-Aldrich) diluted in isopropanol. Staining was next eluted in 100% ethanol and absorbance measured at 500 nm.

### Microscopy and image analysis
Cells were differentiated for 15 days on 12-mm diameter coverslips in 24-well plates. Cells were washed 3 times with PBS before fixation in 4% paraformaldehyde for 10 min. Cells were then incubated in Bodipy (1 μg/ ml, Invitrogen D3922) for 15 min and washed 3 times in PBS before mounting in DAKO Fluorescence Mounting Medium (S3023, Agilent) containing 0.2 μg/ml DAPI. Images were acquired on an IX81 microscope (Olympus) fitted with epifluorescence, a 100× 1.4 NA objective mounted on a piezo drive, and a DeltaVision personalDV (Applied Precision, Ltd.) imaging station. For lipid droplet measurements, images were segmented with ImageJ's Morphological Segmentation plugin MorphoLibJ (https://imagej.net/plugins/morpholibj), and area was measured with the "Analyze particles" function, thresholding on circularity and size.

### Immunoblotting
Proteins were resolved by gradient 4–20% SDS–PAGE, transferred onto nitrocellulose (Cat. 162-0115, BioRad) or PVDF (Cat. IPFL00010, Millipore) membranes and blocked with 5% BSA or non-fat dry milk. Membranes were incubated for 1 h at room temperature or overnight at 4°C using the following antibodies: FAS (Santa Cruz, sc-48357), PPARG (Thermofisher, MA5-14889), CD36 (Santa Cruz, sc-9154), CITED1 (Novus, H00004435-M03), UCP1 (Abcam, 23841), NFIL3 (Abcam, EPR27211-70), γTubulin (Sigma, T5326), Total OXPHOS human antibody cocktail (Abcam, ab11041), and GAPDH (Santa Cruz, sc-25778). Proteins were visualized using IRDye-800- (LI-COR Biosciences, 926-32214), IRDye-680- (LI-COR Biosciences, 926-68023), or HRP-coupled (Jackson ImmunoResearch, 111-035-144 and 115-035-146) secondary antibodies. Bands were quantified by densitometry (BioRad Image Lab) using γTubulin or GAPDH for normalization.

### RNA-seq
Total RNA was isolated from triplicate experiments using the RNAeasy mini kit (Qiagen). Sequencing libraries were prepared using the KAPA mRNA HyperPrep kit (Roche) and sequenced on a Novaseq (Illumina). RNA-seq reads were filtered to remove low-quality reads using fastp v 0.20.1[43]. Filtered reads were aligned to the hg38 genome (GENCODE v32) with hisat2 v2.1.0[44], and counted using featureCounts in Subread v2.0.1 with options -M and --fraction[45]. Low abundance genes were filtered using filterByExpr and then normalized using the trimmed mean of M values (TMM) method from edgeR[46]. Beige and white gene expression was compared between conditions using the robust eBayes method with limma-voom adjustment[47]. DEGs had FDR adjusted *p* value < 0.01 and absolute

log2FC > 1. Heatmaps were generated by hierarchical clustering of expression z-scores using in the R package ComplexHeatmap. To calculate FPKM at promoters, RNA reads were counted 300 bp downstream of each TSS using featureCounts with -O and normalization factors were calculated with the TMM method. Brown vs white adipose tissue logFC were computed for each individual using the highest expressed ensembl transcript for genes with expression > 1 FPKM across all subjects[10]. GSEA using BATLAS brown markers[9], and GO Cellular Component Collagen Containing Extracellular Matrix with clusterProlifer (pvalueCutoff = 1) was used to obtain running enrichment scores. RNA-seq (counts per million) was compared to in vivo adipocyte subtypes using the EPIC deconvolution webtool[48]. The reference cells list was created by aggregating relative counts for marker genes (p value < 0.01, logFC > 0.25 and percentage expressed > 0.25) from each adipocyte subtype.

## ChIP-seq of modified histones

Undifferentiated cells were fixed for 10 min with 1% formaldehyde, lysed in 50 mM Tris-HCl, pH 8, 10 mM EDTA, 1% SDS, protease inhibitors and Na-sodium butyrate, and sonicated in a Bioruptor Pico (Diagenode) into ~200 bp fragments. After sedimentation, the supernatant was diluted 10 times in RIPA buffer and incubated with H3K4me3 (Diagenode c15410003), H3K4me1 (Diagenode c15410037), H3K27ac (Diagenode c15410174), H3K27me3 (Diagenode c15410069), H3K36me3 (Diagenode c15410058) and H3K9me3 (Diagenode c15410056) antibodies (each at 2.5 µg/10⁶ cells) coupled to magnetic Dynabeads Protein A (Invitrogen) for 2 h at 4 °C. ChIP samples were washed, cross-links were reversed and DNA was eluted for 2 h at 68 °C in 50 mM NaCl, 20 mM Tris-HCl pH 7.5, 5 mM EDTA, 1% SDS and 50 ng/µl Proteinase K. DNA was purified using phenol:chloroform:isoamyl alcohol and dissolved in $H_2O$. For D15 differentiated white and beige adipocytes, cells were trypsinized, resuspended in HBSS, 0.5% BSA, and centrifuged 200 g for 5 min to isolate floating mature adipocytes. Purified nuclei were then fixed and processed as cell samples. ChIP-seq libraries were prepared using a Microplex kit (Diagenode) and sequenced with 150-bp paired-end reads on an Illumina NovaSeq.

## ChIP-seq analysis

FASTQ sequences from H3K4me3, H3K27ac, H3K4me1, H3K36me3, H3K27me3 and H3K9me3 ChIPs were aligned to hg38 using Bowtie2 v2.4.5[49]. Peaks were detected using MACS2 v2.2.7.1[50], except for H3K9me3 ChIP where peaks were called using Enriched Domain Detector (http://github.com/CollasLab/edd)[51] with gap penalty and bin size defined as the mean outputs of 10 runs in auto-estimation mode. ChIP-seq read counts were normalized to library size using reads per genome coverage (RPGC) method in deepTools. Log2(ChIP/Input) ratios were calculated using bamCompare in deepTools v3.5.3[52]. Bigwig files of normalized read counts were visualized using Integrative Genomics Viewer[53]. Differential peaks were identified using MAnorm2 v1.2.2 with default parameters[54].

A 15-state chromatin model was generated using chromHMM[55] with aligned reads in 1 kb bins. Six histone modifications each from day0, and day15 white and beige samples were included.

## ATAC-seq

ATAC-seq was done in two independent biological replicates following the Omni-ATAC protocol[56] and optimized for D15 adipocytes. Cells on D0, D1 and D3 were trypsinized and 50,000 viable cells per condition were pelleted at $500 \times g$ for 5 min at 4 °C. Nuclei were isolated by lysing cell pellets in 50 µl of ice-cold RSB buffer (10 mM Tris-HCl, pH 7.4, 10 mM NaCl, 3 mM $MgCl_2$, 0.1% NP40, 0.1% Tween-20, 0.01% digitonin) and ending lysis with 1 ml RSB buffer lacking NP40 and digitonin, followed by centrifugation at $500 \times g$ for 10 min at 4 °C. Supernatants were removed and nuclei pellets were carefully resuspended in 50 µl of transposition mixture (25 µl 2X TD buffer and 2.5 µl transposase (Illumina, 20034197), 16.5 µl PBS, 0.5 µl 1% digitonin, 0.5 µl 10% Tween-20, 5 µl $H_2O$) to carry out the transposition reaction at 37 °C for 30 min on a Thermomixer at 1000 rpm. Cells on D15 were washed with PBS and treated 30 min with 200 U/ml DNAse I (Worthington Biochemical, LS002139) in DMEM/F12 10% FBS at 37 °C. After 4 washes with warm PBS, cells were lysed directly on the culture dish on ice using RSB containing 1% NP40, 0.1% Tween-20 and 0.01% digitonin. After a 5 min centrifugation at $500 \times g$ 4 °C, pelleted nuclei were resuspended in RSB lacking NP40 and digitonin, then stained with Trypan Blue and counted on a Neubauer chamber. 50,000 nuclei were transferred to 1 ml of the same buffer to be pelleted during a 10 min centrifugation at $500 \times g$ 4 °C. Tagmentation was performed as above. Total tagmented DNA was purified using the Zymo DNA Clean and Concentrator-5 kit (Zymo Research, D4014) and eluted in 20 µl elution buffer. Libraries were amplified initially for 5 cycles using dual indexes for tagmented libraries from Diagenode (C01011034) with NEB Next High-Fidelity 2x PCR Master Mix (NEB, M0541). Further amplification was adjusted to the input material as assessed by qPCR using primer pair UDI 1 (Diagenode) for all samples. Library clean-up was done twice with a double-sided selection using AMPure XP beads (Beckman Coulter). Cleaned-up libraries were quantified with the KAPA Library Quantification Kit (Roche, KK4824) and quality assessed using the Agilent TapeStation System, before sequencing at > 200 million read depth with 150 bp paired-end reads on an Illumina NovaSeq.

## ATAC-seq analysis and TF footprinting

ATAC-seq reads were filtered by removing low-quality reads using fastp v0.23.2[43]. Reads were aligned hg38 with Bowtie2 v2.4.5[49]. Duplicates were removed using Picard MarkDuplicates (http://broadinstitute.github.io/picard/) along with mitochondrial reads and reads that aligned with a mapping quality below 10 using samtools v1.9[57]. Genrich v0.6.1 was used to call peaks for each sample (https://github.com/jsh58/Genrich). ATAC differential regions were identified using MAnorm2 v1.2.2 with default parameters[54,58,59]. DARs were considered promoter-associated when located within 2 kb upstream of a TSS harboring a H3K4me3 peak at any differentiation time point. TF footprinting was conducted on merged replicates using TOBIAS v0.16.0[53] with default parameters. JASPAR2022 (CORE_vertebrates_non-redundant) was used for motif enrichment analysis to identify TF binding sites[54]. Normalized read count tracks were generated using deepTools.

## Machine learning model and SHAP analysis

The change in seven epigenetic signal tracks (ATAC-seq and ChIP-seq of H3K4me3, H3K4me1, H3K27ac, H3K27me3, H3K36me3 and H3K9me3) between white and beige were used to predict the average RNA log2 fold change at promoters with a lightGBM model. Epigenetic signals were summarized as a log2 fold change white vs. beige at promoters of DEGs (TSS -2 kb/ + 300 bp) from RPGC normalized read counts. FPKM at promoters was calculated as indicated above.

Python package lightGBM was used to train a gradient boosting regression tree, predicting the expression of 18 282 promoters from seven epigenetic tracks. To determine the optimal hyper parameters, several values of number of estimators, learning rate and the maximum number of leaves per tree were tested with 10-fold cross validation (Supplementary Fig. 7). With fixed parameters for max_bin and min_data_per_leaf (512 and 100 respectively), model accuracy (measured via negative root mean squared error and $r^2$) increased with the number of estimators until about 10, 000 estimators (Supplementary Fig. 7A, B). Accuracy also correlated positively with learning rate. Increasing the maximum number of leaves per estimator increased both model accuracy and computational time (Supplementary Fig. 7B, C) The final parameters to the model were: number of estimators = 10 000, learning rate = 0.05 and the maximum number of leaves per tree = 50 with 10-fold cross validation. A random forest regression model was fitted to the same data (n_estimators = 10, 000, max_leaf_nodes = 50, min_samples_leaf =

100), which took significantly longer to run and produced significantly worse predictions (Supplementary Fig. 7E, F).

## Hi-ChIP and chromatin loop calling

ASCs were differentiated until D15 in two independent biological replicates. Cells were washed twice in warm PBS before lysis in 4 ml Hi-ChIP buffer (10 mM Tris-HCl pH 7.4, 10 mM NaCl, 3 mM MgCl$_2$; 0.01% Digitonin, 0.1% Tween 20 and 1.0% NP-40, 20 mM Na-Butyrate). Lysed cells were centrifugated at 500 g for 5 min at 4 °C. The floating fraction (containing unlysed adipocytes) and the pellet were combined in 5 ml Hi-ChIP buffer and dounced to isolate nuclei. Nuclei were washed twice in Hi-ChIP buffer, followed by centrifugation at $500 \times g$ for 5 min at 4 °C. The nuclei pellet was resuspended in 5 ml Nuclei buffer (10 mM HEPES, 1.5 mM MgCl$_2$, 250 mM sucrose, 0.1% NP-40, 20 mM Na-Butyrate in PBS) and processed for Hi-ChIP according Arima-HiC+ Kit protocol (Cat Nr: A510008), with some adaptations. Briefly, 15 μg DNA equivalent of cross-linked nuclei were lysed in deionized H2O, sonicated for 10 cycles (30 s on, 30 s off; Bioruptor Pico). The ChIP was performed with 2.5 μg H3K27ac antibody (Diagenode C15410174). Libraries were quantified (KAPA, KK4824), amplified (KAPA, KK2620) and barcoded with xGen 2S Plus DNA library prep kit (IDT, 10009877) in combination with xGen 2S MID Adapter (IDT, 10009900). Libraries were sequenced to an average depth of 355 million 150 bp PE reads on an Illumina NovaSeq instrument.

H3K27ac Hi-ChIP was analyzed using MAPS v2.0[14] as implemented within the Arima Genomics pipeline (https://github.com/HuMingLab/MAPS/tree/master/Arima_Genomics). Briefly, the raw paired-end reads were mapped to the hg38 reference genome using bwa mem v0.7.12[57], filtered for uniquely mapped reads, then binned at 5 kb to generate the chromatin contact matrix. MACS2 2.2.9.1 was used for H3K27ac peaks calling[50]. Paired-end reads with at least one end overlapping the H3K27ac ChIP–seq peaks were used to identify long-range chromatin interactions at 5 kb resolution. A positive Poisson model was used for identifying significant interactions (FDR < 0.01).

HiC-CDC+ [15] was used to call differential loops with the following adaption: significant loops with more than 6 read counts in both replicates and a total count over 18 in at least one of two conditions (white and beige) were included.

## Motif enrichment analysis

Motif enrichment analysis was performed on MED1 ChIP peaks[23] overlapping with beige loops anchors against the complete JASPAR database using TFmotifView web server (https://bardet.u-strasbg.fr/tfmotifview/).

## NFIL3 ChIP-PCR

ASCs were differentiated in 10 cm Petri dishes until D15. Nuclei were isolated from approximately $10 \times 10^6$ cells and fixed for 10 min with 1% formaldehyde in Nuclei buffer (10 mM HEPES, 1.5 mM MgCl$_2$, 250 mM sucrose, 0.1% NP-40). Fixed nuclei were lysed in 50 mM Tris-HCl, pH 8, 10 mM EDTA, 1% SDS and protease inhibitors, and sonicated in a Bioruptor Pico (Diagenode) into ~200 bp fragments. After sedimentation, the supernatant was diluted 10 times in RIPA buffer and incubated overnight at 4 °C with 8 μg NFIL3 ChIP-grade antibody (Abcam, EPR27211-70), followed by incubation with magnetic Dynabeads Protein A (Invitrogen) for 2 h at 4 °C. ChIP samples were washed, cross-links were reversed and DNA was eluted for 2 h at 68 °C in 50 mM NaCl, 20 mM Tris-HCl pH 7.5, 5 mM EDTA, 1% SDS and 133 ng/μl Proteinase K. DNA was purified using phenol:chloroform:isoamyl alcohol and dissolved in H$_2$O. ChIP DNA was used as template for quantitative (q)PCR using primers targeting NFIL3 binding sites and a control region[60] (Supplementary Data 5). PCR was done using SYBR Green (BioRad) for 3 min at 95 °C and 40 cycles of 95 °C for 30 s, 60 °C for 30 s, and 72 °C for 20 s.

## siRNA transfection

ASCs were differentiated using the white differentiation protocol, in three independent biological replicates. White adipocytes were transfected with 50 nM siRNAs targeting NFIL3 or a scramble control (MedChemExpress, HY-RS09252) using RNAimax Transfection Reagent (Thermo Fisher, 13778100) according to the manufacturer's protocol. NFIL3 protein knockdown and gene expression were assessed after 72 h.

## Statistics and reproducibility

Statistics were performed on biological replicates, defined as independent differentiation experiments. White and beige differentiation experiments were performed in parallel. Data were plotted as mean ± standard deviation, and the sample size and statistical test for each experiment are indicated in the figure legends. Gaussian distribution of data was tested using the Shapiro-Wilks test prior to parametric test. Cohen's D was used to compare effect sizes of continuous variables in epigenetic data with < 3 biological replicates.

## Reporting summary

Further information on research design is available in the Nature Portfolio Reporting Summary linked to this article.

## Data availability

RNA-seq data, histone ChIPs, ATAC-seq and Hi-ChIP data generated for this study are available at NCBI GEO with accession number GSE293136. Re-analysis of MED1 datasets[23] can be found at GEO under accession GSE256261. RNA-seq from paired white and brown fat biopsies were downloaded from GEO with accession GSE113764 and single nuclei RNAseq was sourced from the Single Cell Portal with accession SCP3116. Uncropped membranes are presented as Supplementary Figs. 10–12. The numerical data underlying the graphs is presented in Supplementary Data 6. All other data are available from the corresponding author on reasonable request.

## Code availability

Code for data processing and analysis is available at Zenodo (https://doi.org/10.5281/zenodo.17625772)[61].

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

## Acknowledgements

We acknowledge the Norwegian Sequencing Centre (Oslo University Hospital) for professional sequencing services. We thank Patrizia Nothnagel for assistance with Seahorse experiment. This work was funded by the University of Oslo, South-East Health Norway grant 40202 to P.C., Research Council of Norway grant 313508 to P.C., and Nansen fund 17368 to N.B.

## Author contributions

S.H.P. and M.A. analyzed data. N.M.G., A.L.S., and J.M.Ø. generated datasets. M.Z. advised on machine learning and statistics. N.B. and S.H.P. made figures. N.B. and P.C. designed the study. N.B. supervised the work. N.B., S.H.P., and P.C. wrote the manuscript. All authors approved the final version of the paper.

## Competing interests

The authors declare no competing interests.
