## [Transparent Peer Review file · Communications Biology]

Multimodal epigenetic and enhancer network remodeling shape the transcriptional landscape of human beige adipocytes

Corresponding Author: Dr Nolwenn briand

Version 0:

Decision Letter:

**** Please ensure you delete the link to your author homepage in this email if you wish to forward it to your coauthors ****

Dear Dr briand,

Your manuscript entitled "Multimodal epigenetic and enhancer network remodeling shape the transcriptional landscape of human beige adipocytes" has now been seen by 4 referees, whose comments are appended below. You will see from their comments below that while they find your work of considerable interest, some important points are raised. We are interested in the possibility of publishing your study in Communications Biology, but would like to consider your response to these concerns in the form of a revised manuscript before we make a final decision on publication.

We therefore invite you to revise and resubmit your manuscript, taking into account the points raised.

Please highlight all changes in the manuscript text file.

We are committed to providing a fair and constructive peer-review process. Do not hesitate to contact us if you wish to discuss the revision in more detail or if there are specific requests from the reviewers that you believe are technically impossible or unlikely to yield a meaningful outcome.

At the same time, we ask that you ensure your manuscript complies with our editorial policies. Specifically:

For all graphs depicting a single point value (e.g., mean) with error bars, you must add individual data points or convert the graph to a boxplot or dot-plot to show data distribution.

It's mandatory to provide access to the numerical source data for graphs and charts either through a repository or by providing the data in a Supplementary Data file (in excel format).

All blots/gels must be accompanied by size markers in every figure panel. Uncropped and unedited blot/gel images must be included as Supplementary Figure(s) in the Supplementary Information pdf.

Please ensure that you have complied with the data deposition policies at the Nature Portfolio, please see here.

Please ensure that you have complied with our policies on research involving animals and humans, see here

Please follow the ARRIVE guidelines for reporting animal experiments. Please fully complete an ARRIVE checklist including both the essential and recommended set of items (adding information to the manuscript where needed) and upload this with your revised manuscript.

Please also see [our revision checklist](https://www.nature.com/documents/CommsBio-file-checklist-revision.pdf) for guidance on formatting the manuscript and complying with our policies. A comprehensive guide to our formatting requirements for final submissions is also available for your reference [here](https://www.nature.com/documents/commsj-life-style-formatting-guide-accept.pdf).

Please use the following link to submit your revised manuscript, point-by-point response to the referees' comments (which should be in a separate document to the cover letter) and any additional files:

Link Redacted

When submitting the revised version of your manuscript, please pay close attention to our [Digital Image Integrity Guidelines](https://www.nature.com/commsbio/editorial-policies/image-integrity).

We would like to receive your revision within 4 weeks, but appreciate that every situation is unique. We look forward to receiving your revised manuscript when it is ready, and will not enforce a hard deadline on this revision.

Please do not hesitate to contact me if you have any questions or would like to discuss these revisions further. We look forward to seeing the revised manuscript and thank you for the opportunity to review your work.

Best regards,

Simona Chera, PhD
Editorial Board Member
Communications Biology
orcid.org/0000-0001-6310-3486

Referee expertise:

Referee #1: Molecular Metabolism, Spatial Genomics, Single Cell Genomics

Referee #2: Physiology, molecular mechanisms of energy metabolism

Referee #3: Metabolism

Reviewers' comments:

Reviewer #1 (Remarks to the Author):

The manuscript titled "Multimodal epigenetic and enhancer network remodeling shape the transcriptional landscape of human beige adipocytes" presents a comprehensive multi-omics analysis of human adipocyte differentiation, comparing the development of "white" and thermogenic "beige" adipocytes. By integrating transcriptomic, epigenomic (histone modifications and chromatin accessibility), and three-dimensional chromatin-interaction (Hi-ChIP) data—together with machine-learning approaches—the authors dissect the regulatory mechanisms underlying adipocyte "beiging." The authors offer novel insights into the distinct epigenetic programs of rosiglitazone-induced cells over "white" adipocytes. They show that gene expression of "white" adipocytes correlates with H3K27ac activation at enhancers and promoters, whereas rosiglitazone-induced cells uniquely gain H3K4me3 at promoters of mitochondrial genes. They also reveal extensive enhancer-network remodeling during rosiglitazone induction, including new short-range enhancer loops near fatty-oxidation and thermogenic genes, and they identify NFIL3 as a putative suppressor of the rosiglitazone induced gene program. Overall, the data appear robust and the manuscript is well organized. This work advances our understanding of the chromatin dynamics and regulatory networks that define rosiglitazone-induced cells.

While the manuscript is strong, there are a few points that could be clarified or addressed to further improve its quality and impact:

A. The manuscript (Fig. 1E–I) relies heavily on the rise in UCP1 mRNA to validate a beige phenotype. Recent human studies show that a large increase in UCP1 transcripts does not always translate into functional beige adipocytes (PMID: 23353596). For example, BMP4 or BMP7 treatments up-regulate UCP1 mRNA without detectable protein or respiration, illustrating that mRNA alone is insufficient (PMID: 28425843). Please consider adding protein-level and functional validation: (1) Western blot or immunofluorescence for UCP1 and ETC complexes at the same time-points used for RNA-seq/ATAC-seq. (2) Respirometry (Seahorse or equivalent) under (i) basal conditions, (ii) rosiglitazone alone, and (iii) acute β -adrenergic stimulation (e.g., CL-316, 243 or mirabegron).

B. How do these rosiglitazone-induced cells and "white" adipocytes represent native human beige and white adipocytes?

The authors may consider leveraging publicly available bulk or single-cell RNA-seq datasets from human beige/brown and white adipocytes to compare how much of the gene-expression program is conserved between rosiglitazone-induced cells and native beige cells, as well as white adipocytes.

Reviewer #2 (Remarks to the Author):

Pickering and colleagues describe differences in the molecular remodeling which associates with human white vs beige adipogenesis. While the applied analyses are mostly sound and well performed, a weakness of this study arises from the cellular model which has been used. The adipocytes have been in vitro differentiated from ASCs of a single female donor, and could have been backed up with primary human material from a number of patients. As a hypothesis-generating study, this paper has some interesting value, as it suggests various molecular changes occurring during human beige adipogenesis. Concerning the interpretation of the results, the value of the paper could also be considerably increased by applying some causality in their project by, e.g., knocking down NFIL3 or mutating the PDK4 promoter.

Minor: line 248, typo PDK4

Reviewer #3 (Remarks to the Author):

Summary

This study explores the epigenetic and genetic networks that are changed as a function of human adipocyte differentiation. Through integrative transcriptomic, epigenomic, and 3D genome analyses, the researchers show that adipocyte fate (white vs. beige) is tightly regulated by distinct epigenetic signatures. White adipocyte gene expression is primarily controlled by promoter H3K27ac levels and chromatin accessibility, while beige adipocyte-specific mitochondrial genes are governed by promoter H3K4me3 remodeling. Beige adipogenesis is marked by a late and pronounced increase in chromatin accessibility at both enhancers and promoters. This is accompanied by a targeted reorganization of the 3D genome, with increased recruitment of short-range enhancers that regulate genes involved in fatty acid oxidation and thermogenesis. These new enhancer-promoter contacts coincide with chromatin opening at C/EBP transcription factor binding sites. The study also identifies NFIL3 as a negative regulator of the beige program, binding to enhancers of the PDK4 gene, which is crucial for fatty acid oxidation. Overall, the findings highlight that the establishment of beige adipocyte identity is driven by a complex interplay of chromatin state transitions, enhancer rewiring, and transcription factor dynamics. This multimodal regulation underpins the unique metabolic properties of beige adipocytes and underscores their therapeutic potential for metabolic diseases.

Major Points

My major point kind of centers on the last experiment – the ChIP PCR of NFIL3 looking to find it on the enhancer regions associated with PDK4. First, I think you need an IgG control for this experiment to show the amount of DNA that you're precipitating in the background. But more importantly, I think it could really enhance the manuscript if this assay was performed on cells where you changed the Rosiglitazone regimen. That's really my major point – that this manuscript uses the phrase "beige adipocyte" interchangeably with "rosi treated white adipocyte." I want to know which of the changes that they report, either to the DNA, histones, loops, or binding protein occupancy, are associated with changes in cellular identity and which are acutely responsive to Rosi. The authors state that beige adipocytes revert to a white phenotype in vivo, so would they expect that removing rosi from their beige cells would recruit more NFIL3 to the PDK enhancer? Would adding rosi in the final 24 hours to white cells recapitulate any of the differences they saw in "beige" cells compared to white cells? Especially since they report the major transcriptional differences are mostly appearing in the final 3 days of differentiation, it would really help to know how permanent the changes they see are, or likewise how rapidly they can respond to ppargamma agonism.

In Figure 1, panels C and I, the western blot quantification seems to have been normalized such that the replicate control samples are all set to a value of 1. Since this leads to a non-normal distribution in one of the groups, it seems the authors have chosen to test the statistical significance of the difference between white and beige cells with a Wilcoxon signed ranked test, however it's unclear why all of the data needs to be normalized this way to begin with. Can't the western data for each of the three replicate experiments be normalized to the average densitometry values for the control samples? That would allow the reader to get a sense of the variance inherent to this assay. It's also not entirely clear to me that the data meets the other assumptions of the Wilcoxon test, for example how do the authors see the white and beige conditions as paired data? Maybe because they are cells from the same person, but they've been treated separately so this seems a bit of a stretch. This happens in supplemental 9d too.

The sentence that starts in line 35 is a bit ambiguous, and would make more sense like this (notice the removal of the comma): "We notably uncover NFIL3 as [a] negative regulator of the beige program differentially bound at enhancers of PDK4, a key metabolic switch towards fatty acid oxidation."

I like the way the heatmap in Figure 2D is labeled on the bottom with the White and Beige and the arrows. Can you label the heatmap in Figure 1F the same way for consistency?

On line 89, the phrase "mitochondrial proteome" struck me a bit. I guess you're setting yourself up for the western blots with the OXPHOS cocktail, but do any of the genes that you're blotting for with this cocktail show differential regulation in the RNA sequencing data, or is the term "Mitochondrial Matrix" showing up in your gene ontology analysis driven by other genes

(e.g. transcriptional regulators of OXPHOS genes)? It could be helpful to include the mRNA expression of the genes in the OXPHOS cocktail in a supplemental figure.

Line 106-108 was confusing. Is the "...chromatin state model learned from our ChIP seq data..." the thing that you developed with the all of your different ChIP assays (in the methods it says you did ChIP seq for H3K27ac, H3K4me3, H3K4me1, H3K27ac, H3K27me3, H3K36me3, and H3K9me3)? I think it might be helpful to say that these were all done in the results before you say that the DARs map to chromatin state regions defined by these experiments. Also in this sentence, if DARs is plural, shouldn't it be "annotate" rather than "annotates"?

Line 163 I don't think that regulations should be plural.

Line 173 I wasn't clear how the samples were considered biological replicates. It looks like there are two samples labeled "W1" and "W2" and two samples "B1" and "B2" and in the methods they say Hi-ChIP started with 10cm dishes. This sounds like they have technical replicates rather than biological replicates.

It would have been excellent to do the C/EBP ChIP PCR similar to what was done with NFIL3. Wonder why this wasn't done?

Reviewer #4 (Remarks to the Author):

I co-reviewed this manuscript with one of the reviewers who provided the listed reports. This is part of the Communications Biology initiative to facilitate training in peer review and to provide appropriate recognition for Early Career Researchers who co-review manuscripts.

** See the Nature Portfolio author and referees' website at www.nature.com/authors for information about policies, services and author benefits

Communications Biology is committed to improving transparency in authorship. As part of our efforts in this direction, we are now requesting that all authors identified as 'corresponding author' create and link their Open Researcher and Contributor Identifier (ORCID) with their account on the Manuscript Tracking System prior to acceptance. ORCID helps the scientific community achieve unambiguous attribution of all scholarly contributions. You can create and link your ORCID from the home page of the Manuscript Tracking System by clicking on 'Modify my Springer Nature account' and following the instructions in the link below. Please also inform all co-authors that they can add their ORCIDs to their accounts and that they must do so prior to acceptance.

Version 1:

Decision Letter:

** Please ensure you delete the link to your author homepage in this email if you wish to forward it to your coauthors **

Dear Dr briand,

Your manuscript entitled "Multimodal epigenetic and enhancer network remodeling shape the transcriptional landscape of human beige adipocytes" has now been seen by 4 referees, whose comments are appended below. You will see from their comments below that while they find your work of considerable interest, some important points are raised. We are interested in the possibility of publishing your study in Communications Biology, but would like to consider your response to these concerns in the form of a revised manuscript before we make a final decision on publication.

We therefore invite you to revise and resubmit your manuscript, taking into account the points raised. In particular:

- please address the remaining concerns from Reviewer #3
- please add the Reviewer only Figures as Supplemental data

Please highlight all changes in the manuscript text file.

We are committed to providing a fair and constructive peer-review process. Do not hesitate to contact us if you wish to

discuss the revision in more detail or if there are specific requests from the reviewers that you believe are technically impossible or unlikely to yield a meaningful outcome.

At the same time, we ask that you ensure your manuscript complies with our editorial policies. Specifically:

For all graphs depicting a single point value (e.g., mean) with error bars, you must add individual data points or convert the graph to a boxplot or dot-plot to show data distribution.

It's mandatory to provide access to the numerical source data for graphs and charts either through a repository or by providing the data in a Supplementary Data file (in excel format).

All blots/gels must be accompanied by size markers in every figure panel. Uncropped and unedited blot/gel images must be included as Supplementary Figure(s) in the Supplementary Information pdf.

Please ensure that you have complied with the data deposition policies at the Nature Portfolio, please see [here](http://www.nature.com/authors/policies/availability.html#data).

Please ensure that you have complied with our policies on research involving animals and humans, see [here](https://www.nature.com/commsbio/editorial-policies/ethics-and-biosecurity#studies-involving-animals-and-human-research-participants)

Please follow the ARRIVE guidelines for reporting animal experiments. Please fully complete an [ARRIVE checklist](https://arriveguidelines.org/sites/arrive/files/documents/Author%20Checklist%20-%20Full.pdf) including both the essential and recommended set of items (adding information to the manuscript where needed) and upload this with your revised manuscript.

Please also see [our revision checklist](https://www.nature.com/documents/CommsBio-file-checklist-revision.pdf) for guidance on formatting the manuscript and complying with our policies. A comprehensive guide to our formatting requirements for final submissions is also available for your reference [here](https://www.nature.com/documents/commsj-life-style-formatting-guide-accept.pdf).

Please use the following link to submit your revised manuscript, point-by-point response to the referees' comments (which should be in a separate document to the cover letter) and any additional files:

Link Redacted

When submitting the revised version of your manuscript, please pay close attention to our [Digital Image Integrity Guidelines](https://www.nature.com/commsbio/editorial-policies/image-integrity).

We would like to receive your revision within 4 weeks, but appreciate that every situation is unique. We look forward to receiving your revised manuscript when it is ready, and will not enforce a hard deadline on this revision.

Please do not hesitate to contact me if you have any questions or would like to discuss these revisions further. We look forward to seeing the revised manuscript and thank you for the opportunity to review your work.

Best regards,

Simona Chera, PhD
Editorial Board Member
Communications Biology
orcid.org/0000-0001-6310-3486

Reviewers' comments:

Reviewer #1 (Remarks to the Author):

The authors addressed all my questions, I do not have further concern.

Reviewer #2 (Remarks to the Author):

all concerns have been addressed appropriately.

Reviewer #3 (Remarks to the Author):

I concede that IgG isn't a perfect negative control for the NFIL3 chip, but I don't really think the gene desert control is a perfect negative control either. First of all, it's completely unclear where in the genome this primer set is targeted to, and what if these primers don't produce any amplicon at all? I think an absolute minimum for this experiment is to better describe what these control primers are designed to target, preferably with a reference or experimental evidence that they do in fact produce a 116bp amplicon from unprecipitated genomic DNA. Also, if the authors are trying to make the point that there is locus specific binding to enhancer 1 and not enhancer 2 of the PDK gene, shouldn't there be a statistical analysis of these differences? I think it's obvious that there's on average more DNA pulled down from the intronic region and enhancer 1 than there is for enhancer 2, but for the beige cells the binding to enhancer 1 has a relatively large amount of variability, so how do you conclude there is significantly more binding to enh1 than enh2? Lastly, in the reviewer only figure 6 the authors did the thing where they set all the control samples to 1, so there's no variance in the white samples for any of the 3 genes. That's not possible. I get that it's for reviewers only and they don't specify what test led them to annotate these differences as significant, but I don't understand how they could come to the conclusion that qPCR gives the same value for technical replicates of the same sample.

Reviewer #4 (Remarks to the Author):

I co-reviewed this manuscript with one of the reviewers who provided the listed reports. This is part of the Communications Biology initiative to facilitate training in peer review and to provide appropriate recognition for Early Career Researchers who co-review manuscripts.

** See the Nature Portfolio author and referees' website at www.nature.com/authors for information about policies, services and author benefits

Communications Biology is committed to improving transparency in authorship. As part of our efforts in this direction, we are now requesting that all authors identified as 'corresponding author' create and link their Open Researcher and Contributor Identifier (ORCID) with their account on the Manuscript Tracking System prior to acceptance. ORCID helps the scientific community achieve unambiguous attribution of all scholarly contributions. You can create and link your ORCID from the home page of the Manuscript Tracking System by clicking on 'Modify my Springer Nature account' and following the instructions in the link below. Please also inform all co-authors that they can add their ORCIDs to their accounts and that they must do so prior to acceptance.

Version 2:

Decision Letter:

*** REMEMBER TO ATTACH AIP TABLE ***

** Please ensure you delete the link to your author homepage in this email if you wish to forward it to your coauthors **

Dear Dr Briand,

Your manuscript entitled "Multimodal epigenetic and enhancer network remodeling shape the transcriptional landscape of human beige adipocytes" has now been seen again by our referees, whose comments appear below. In light of their advice I am delighted to say that we are happy, in principle, to publish a suitably revised version in Communications Biology.

We therefore invite you to revise your paper one last time to address the remaining concerns of our reviewers. At the same time we ask that you edit your manuscript to comply with our format requirements and to maximise the accessibility and therefore the impact of your work.

Please add the Code to the Zenodo (not github) and cite properly.

Please make sure the margins/sides of the uncropped gel images in Supplementary are visible.

Add title and Author information in the Supplementary files.

Please make sure to check the attached AIP Table (CommsBio AIP table COMMSBIO-25-4638B.docx file) minutely, with special attention to the highlighted field.

* Please see the attached document for editorial requests for the final version (CommsBio AIP table COMMSBIO-25-4638B.docx file). Please ensure a completed version of this file is uploaded as a Related Manuscript with your final submission.

* Please review our [final submission file checklist](https://www.nature.com/documents/commsj-file-checklist.pdf) to ensure all necessary files are present with your final submission and to avoid delays in accepting your manuscript. For your reference, a style and formatting guide is available [here](https://www.nature.com/documents/commsj-life-style-formatting-guide-accept.pdf) and includes all of our style requirements.

It is important that you pay careful attention to the requests in these documents to avoid a delay in formal acceptance of the article.

Open access

Communications Biology is a fully open access journal. Articles are made freely accessible on publication. For further information about article processing charges, open access funding, and advice and support from Nature Research, please visit <https://www.nature.com/commsbio/open-access>

Please use the following link to upload your revised files:

Link Redacted

We hope to hear from you within two weeks. If you expect the process to take longer than one month, please let us know.

Congratulations on an excellent paper!

Best regards,

Simona Chera, PhD
Editorial Board Member
Communications Biology
orcid.org/0000-0001-6310-3486

and

Nilanjan Banerjee, PhD
Associate Editor
Communications Biology

PS: At acceptance, the corresponding author will be provided with instructions for completing the license on behalf of all authors. This grants us the necessary permissions to publish your paper. Additionally, you will be asked to declare that all required third party permissions have been obtained, and to provide billing information in order to pay the article-processing charge (APC).

REVIEWERS' COMMENTS:

Reviewer #3 (Remarks to the Author):

I am satisfied with the responses to my comments and I think that the manuscript should be published in its current form.

** See the Nature Portfolio author and referees' website at www.nature.com/authors for information about policies, services and author benefits

Reviewer #1 (Remarks to the Author):

The manuscript titled “Multimodal epigenetic and enhancer network remodeling shape the transcriptional landscape of human beige adipocytes” presents a comprehensive multi-omics analysis of human adipocyte differentiation, comparing the development of “white” and thermogenic “beige” adipocytes. By integrating transcriptomic, epigenomic (histone modifications and chromatin accessibility), and three-dimensional chromatin-interaction (Hi-ChIP) data— together with machine-learning approaches—the authors dissect the regulatory mechanisms underlying adipocyte “beiging.”

The authors offer novel insights into the distinct epigenetic programs of rosiglitazone-induced cells over “white” adipocytes. They show that gene expression of “white” adipocytes correlates with H3K27ac activation at enhancers and promoters, whereas rosiglitazone-induced cells uniquely gain H3K4me3 at promoters of mitochondrial genes. They also reveal extensive enhancer-network remodeling during rosiglitazone induction, including new short-range enhancer loops near fatty-oxidation and thermogenic genes, and they identify NFIL3 as a putative suppressor of the rosiglitazone induced gene program. Overall, the data appear robust and the manuscript is well organized. This work advances our understanding of the chromatin dynamics and regulatory networks that define rosiglitazone-induced cells.

We thank the reviewer for the positive assessment of our work.

While the manuscript is strong, there are a few points that could be clarified or addressed to further improve its quality and impact:

A. The manuscript (Fig. 1E–I) relies heavily on the rise in UCP1 mRNA to validate a beige phenotype. Recent human studies show that a large increase in UCP1 transcripts does not always translate into functional beige adipocytes (PMID: 23353596). For example, BMP4 or BMP7 treatments up-regulate UCP1 mRNA without detectable protein or respiration, illustrating that mRNA alone is insufficient (PMID: 28425843). Please consider adding protein-level and functional validation:

(1) Western blot or immunofluorescence for UCP1 and ETC complexes at the same time-points used for RNA-seq/ATAC-seq.

We agree that UCP1 mRNA levels alone are not sufficient to define the beige adipocyte phenotype. In our study, we have already addressed this by including protein-level validation of both UCP1 (Figure 1B,C) and electron transport chain (OXPHOS) complexes (Figure 1H,I). We confirm that rosiglitazone-treated (beige Day 15) adipocytes express UCP1 protein and display higher levels of OXPHOS complexes I-IV, supporting a beige phenotype of mature adipocytes. However, based on our RNAseq data, beige-specific markers are not yet expressed or differentially regulated at earlier time points (Day 1 and Day 3 after differentiation induction) (Reviewer Fig. 1). Therefore, we focused our protein-level validation on Day 15, where transcriptomic data show a robust induction of the beige program, and functional proteins are expressed.

Reviewer Fig. 1: Expression level (FPKM) of beige adipocyte markers during the differentiation time-course

(2) Respirometry (Seahorse or equivalent) under (i) basal conditions, (ii) rosiglitazone alone, and (iii) acute β -adrenergic stimulation (e.g., CL-316, 243 or mirabegron).

We have done a respirometry assay (**Reviewer Fig.2**, also see revised **Fig.1J,K**). We find marked differences in the basal, proton leak, forskolin-induced and maximal OCR in beige compared to white adipocytes, as well as increased spare respiratory capacity. These results are consistent with the previously described metabolic reprogramming of human adipocytes by rosiglitazone treatment (Lee et al. 2019; PMID: 30782959).

Reviewer Fig. 2: Real-time oxygen consumption rates (OCR) of white and beige adipocytes. **A** OCR traces and **B** quantification of respiratory profiles. The following drugs were added sequentially: (i) 2 μ M Oligomycin (ii) 10 μ M forskolin, (iii) 2.5 μ M BAM15 and (iv) 1 μ M Rotenone/Antimycin ($*p < 0.05$, $***p < 0.001$; Two-tailed unpaired Student's T test).

The results and methods sections were edited as follow:

Results:

Line 98: "Functionally, measurements of oxygen consumption rates confirm higher basal, maximal, uncoupled and forskolin-induced respiration in beige adipocytes (Fig.1J, K)."

Methods:

Line 360: "Oxygen consumption. Real-time oxygen consumption rates (OCR) of white and beige adipocytes were measured in XF media (non-buffered DMEM containing 2mM L-glutamine, 25mM glucose, 1mM sodium pyruvate and 2% BSA) using a Seahorse XFe 96 Analyzer (Agilent Technologies). During the assay, 2 μ M Oligomycin (O4876, Sigma), 10 μ M forskolin, 2.5 μ M BAM15 (HY-110284, MedChemExpress), and 1 μ M Rotenone/Antimycin (Agilent) were added sequentially. The quantifications presented are corrected for non-mitochondrial respiration."

B. How do these rosiglitazone-induced cells and "white" adipocytes represent native human beige and white adipocytes? The authors may consider leveraging publicly available bulk or single-cell RNA-seq datasets from human beige/brown and white adipocytes to compare how much of the gene-expression program is conserved between rosiglitazone-induced cells and native beige cells, as well as white adipocytes.

We thank the reviewer for the suggestion but would like to point out that transcriptomic data for a purified population of human subcutaneous beige adipocytes are not yet publicly available. To circumvent this limitation, we used two complementary approaches:

(i) We have performed a gene set enrichment analysis (GSEA) using the BATLAS gene set (Perdikari Cell Reports 2018) as a beige signature, and the GO term Cellular Component Collagen Containing Extracellular Matrix gene sets (GO:0062023) as a white signature. When compared to previously published RNAseq of paired BAT and WAT from supraclavicular biopsies in 14 individuals (Din et al. 2018; PMID: 29909972; GSE113764), our in vitro differentiated beige adipocytes yield highly similar enrichment profiles, with a strong enrichment of BATLAS "brown genes" in both BAT and our beige adipocytes (**Reviewer Fig. 3A**; revised **Fig. S2C**). Conversely, the human WAT vs BAT signature shows an enrichment for ECM-related genes, consistent with our in vitro data (see **Fig. 1G**).

(ii) We have leveraged the recently published snRNAseq dataset (Miranda et al. Nature 2025), and deconvoluted our bulk RNAseq data on the mature adipocytes clusters defined in that study (see Material and Methods – RNA-seq). We find that the majority of our white adipocytes resemble a common subcutaneous adipocyte subtype defined by expression of lipid droplet and housekeeping genes (AD3; **Reviewer Fig. 3B**; revised **Fig. S2D**). By contrast, our beige adipocyte transcriptome matches that of adipocytes enriched for creatine thermogenesis, lipolysis and TCA cycle pathways (AD2; **Reviewer Fig. 3B**; revised **Fig. S2D**).

Notably, no adipocyte cluster enriched for canonical UCP1-dependent thermogenesis was identified in the subcutaneous abdominal adipose tissue profiled by Miranda et al., so that our UCP1-positive population cannot be detected by this deconvolution approach. Taken together, these results indicate that our *in vitro* beige adipocytes represent cell states compatible with both canonical and non-canonical thermogenic programs, underscoring their relevance as a model for human beige adipocyte biology (Wang et al., Cell Metabolism 2024).

Altogether, these analyses support the relevance of our rosiglitazone-induced beige adipocytes as a model for human thermogenic adipocytes and confirm that the molecular differences we describe are consistent with native human adipose cell states.

Reviewer Fig. 3: A Gene set enrichment analysis of paired human BAT vs WAT and Beige vs white *in vitro* differentiated adipocytes using BATLAS brown markers and GOCC “Collagen ECM” as signature for beige and white adipocytes, respectively. **B** Estimated abundance of *in vivo* adipocyte subtypes (Miranda et al. 2025) in day 15 (D15) white and beige adipocytes based on deconvolution of RNA-seq (n = 3 independent differentiations; * p < 0.05, *** p < 0.001, t-test).

The results and methods sections were edited as follow:

Results:

Line 101: “To assess the physiological relevance of *in vitro* beige adipocytes, we compared their transcriptional profiles with human adipose tissues. The transcriptional differences between beige and white adipocytes, including enrichment for the BATLAS brown gene signature⁹ and depletion of extracellular matrix genes, mirror those observed between human brown and white adipose tissues¹⁰ (Supplemental Fig. S2C). Furthermore, RNA-seq deconvolution using a recent snRNA-seq atlas of human adipocytes¹¹ reveals that white and beige adipocytes map to distinct adipose subpopulations, reflecting a shift from lipid storing to thermogenic functions (Supplemental Fig. S2D).”

Methods:

Line 410: Brown vs white adipose tissue logFC were computed for each individual using the highest expressed ensemble transcript for genes with expression > 1 FPKM across all subjects¹⁰. GSEA using BATLAS brown markers⁹, and GO Cellular Component Collagen Containing Extracellular Matrix with clusterProlifer (pvalueCutoff =1) was used to obtain running enrichment scores. RNA-seq (counts per million) was compared to *in vivo* adipocyte subtypes using the EPIC deconvolution webtool⁴⁷. The reference cells list was created by aggregating relative counts for marker genes (p value < 0.01, logFC > 0.25 and percentage expressed > 0.25) from each adipocyte subtype.

Reviewer #2 (Remarks to the Author):

1. Pickering and colleagues describe differences in the molecular remodeling which associates with human white vs beige adipogenesis. While the applied analyses are mostly sound and well performed, a weakness of this study arises from the cellular model which has been used. The adipocytes have been *in vitro* differentiated from ASCs of a single female donor, and could have been backed up with primary human material from a number of patients.

The strength of the model lies in the parallel white and beige differentiation, on the same genetic background, followed by extensive omics characterization. We have recently published parallel white vs beige differentiation from unrelated 6 donors (Hazell Pickering et al. 2024; PMID: 38859907; GSE256262), so the transcriptional response to Rosiglitazone-induced beigeing is not specific to the donor used here. We show here for the reviewer the massive transcriptional upregulation of *CD36*, *FABP4*, *LPL* and *PDK4* genes – which loci display increased chromatin contacts in beige-differentiated cells - in beige vs white adipocytes for all donors (**Reviewer Fig. 2**).

Reviewer Fig. 4: Expression level (FPKM) of genes with increased H3K27ac Hi-ChIP loops (see **Figure 4I**) in beige adipocyte in white and beige adipocytes from 6 independent subjects (S1-6). ChIP-seq, ATAC-seq and Hi-ChIP experiments were performed in **S6** adipocytes.

In addition, using gene set enrichment analysis (GSEA), we now show that our in vitro beige and white adipocytes have comparable upregulation of white and beige transcripts to paired biopsies from human BAT and WAT (see **Reviewer Fig.3A**; new **Fig. S2C**). Similarly, deconvolution from snRNAseq shows our cell types overlap with subpopulations of adipocytes found in human WAT, with an increase of thermogenic adipocytes in beige compared to white adipocytes (see **Reviewer Fig.3B**; new **Fig. S2D**). Thus, our in vitro model recapitulates features of white and beige human in vivo adipocytes (also see response to Reviewer #1 – B).

2. As a hypothesis-generating study, this paper has some interesting value, as it suggests various molecular changes occurring during human beige adipogenesis. Concerning the interpretation of the results, the value of the paper could also be considerably increased by applying some causality in their project by, e.g., knocking down NFIL3 or mutating the PDK4 promoter.

We agree that perturbation of transcription factor expression can in some settings provide valuable mechanistic insights. To address this point, we transiently knocked down NFIL3 in D15 white and beige adipocytes using 3 different siRNAs. Adipocytes are inherently difficult to transfect, and in our hands all 3 siRNA yield a reproducible 60% reduction of NFIL3 protein level (**Reviewer Fig. 5A, B**). However, this global knock down does not affect *PDK4* expression (**Reviewer Fig. 5C**). Several factors may explain this apparent lack of effect: (i) the DNA-bound fraction of NFIL3 may have a lower turnover, so that partial depletion is insufficient to reduce occupancy; (ii) NFIL3 functions as part of the PAR-bZIP network (with DBP, TEF, and HLF), such that partial loss may be buffered by redundant factors; and (iii) enhancer redundancy is common in mammalian gene regulation (Osterwalder et al., PMID 29420474), and indeed, *PDK4* is engaged by multiple distal enhancers (Fig. S10), some of which are not bound by NFIL3. We therefore interpret this negative result with caution: it does not exclude a potential repressive effect of NFIL3 binding at enhancer or intronic

sites. Importantly, our ChIP-qPCR data demonstrate NFIL3 occupancy at two distinct loci near PDK4, with reduced binding in beige adipocytes, supporting a model in which the recruitment of other C/EBP family members contributes to PDK4 activation during beige differentiation.

Reviewer Fig. 5: Western blot analysis (A) and quantification (B) of NFIL3 protein expression levels in white adipocytes 72h after transfection with one Scramble (siSCR) and 3 NFIL3-specific siRNAs (n = 3 independent differentiations; *** p < 0.0001, one way ANOVA). (C) RT-qPCR analysis of *PDK4* gene expression in siSCR and siNFIL3 transfected white adipocytes. NT: non-transfected.

Regarding mutating the PDK4 promoter: we would like to clarify that there is no NFIL3 binding site at the PDK4 promoter; our footprinting analysis highlights some of the PDK4 *enhancers* as a putative target for NFIL3 (see **supplemental Fig. S10**), and we show NFIL3 binding at one enhancer and the contacted intronic region (**Fig. 5G**). For the footprinting analysis, the identity of differentially bound transcription factors is inferred from transcription factor binding motifs. The NFIL3 and CEBPs motifs are highly redundant (**Fig. 5C**), leading to a large overlap of their footprints (**supplemental Fig. S10**). Indeed, NFIL3 has previously been shown to compete with C/EBPs for DNA binding (Li, F et al. Mol Endocrinol 2011; MacGillavry, H. D. et al. Mol Cell Neurosci 2011). Thus, mutating the NFIL3 binding site would most likely also alter the binding of other CEBPs, making the experiment inconclusive regarding NFIL3 specifically.

3. Minor: line 248, typo PDK4
Corrected

Reviewer #3 (Remarks to the Author):

Summary

This study explores the epigenetic and genetic networks that are changed as a function of human adipocyte differentiation. Through integrative transcriptomic, epigenomic, and 3D genome analyses, the researchers show that adipocyte fate (white vs. beige) is tightly regulated by distinct epigenetic signatures. White adipocyte gene expression is primarily controlled by promoter H3K27ac levels and chromatin accessibility, while beige adipocyte-specific mitochondrial genes are governed by promoter H3K4me3 remodeling. Beige adipogenesis is marked by a late and pronounced increase in chromatin accessibility at both enhancers and promoters. This is accompanied by a targeted reorganization of the 3D genome, with increased recruitment of short-range enhancers that regulate genes involved in fatty acid oxidation and thermogenesis. These new enhancer-promoter contacts coincide with chromatin opening at C/EBP transcription factor binding sites. The study also identifies NFIL3 as a negative regulator of the beige program, binding to enhancers of the PDK4 gene, which is crucial for fatty acid oxidation. Overall, the findings highlight that the establishment of beige adipocyte identity is driven by a complex interplay of chromatin state transitions, enhancer rewiring, and transcription factor dynamics. This multimodal regulation underpins the unique metabolic properties of beige adipocytes and underscores their therapeutic potential for metabolic diseases.

Major Points

My major point kind of centers on the last experiment – the ChIP PCR of NFIL3 looking to find it on the enhancer regions associated with PDK4.

1. First, I think you need an IgG control for this experiment to show the amount of DNA that you're precipitating in the background.

IgG controls were not included in our ChIP-PCR experiments because they typically yield minimal and highly variable (non-reproducible) DNA pulldown and do not necessarily reflect the non-specific background of the NFIL3 antibody. Instead, to rigorously assess ChIP specificity, we included multiple internal negative controls such as (i) a heterochromatic "gene desert" region where we do not expect NFIL3 to bind, and (ii) an unbound region within the *PDK4* locus. (iii) We also included a region that is equally bound by NFIL3 in both white and beige adipocytes, arguing for the specificity of the beige-response we report (reduced NFIL3 binding in beige adipocytes; **Fig. 5G**). These controls provide a locus- and antibody-specific assessment of the background signal.

2. But more importantly, I think it could really enhance the manuscript if this assay was performed on cells where you changed the Rosiglitazone regimen. That's really my major point – that this manuscript uses the phrase "beige adipocyte" interchangeably with "rosi treated white adipocyte." I want to know which of the changes that they report, either to the DNA, histones, loops, or binding protein occupancy, are associated with changes in cellular identity and which are acutely responsive to Rosi. The authors state that beige adipocytes revert to a white phenotype in vivo, so would they expect that removing rosi from their beige cells would recruit more NFIL3 to the PDK enhancer? Would adding rosi in the final 24 hours to white cells recapitulate any of the differences they saw in "beige" cells compared to white cells? Especially since they report the major transcriptional differences are mostly appearing in the final 3 days of differentiation, it would really help to know how permanent the changes they see are, or likewise how rapidly they can respond to ppargamma agonism.

We would like to clarify that in our transcriptomic time course including day 0, 1, 3 and 15 of differentiation, we did not observe major differences between white and rosiglitazone-treated adipocytes at the early time points (day 1 and day 3); significant transcriptional divergence became apparent only at day 15.

To address whether this reflects a change in cellular identity, we directly tested the effect of rosiglitazone exposure/removal at the end of differentiation. Specifically, we (i) differentiated white adipocytes with Rosi added only during the last 24 hours, and (ii) differentiated beige adipocytes with Rosi withdrawn for the final 24 hours. We find that although *PDK4* expression is rapidly induced by Rosi in white adipocytes, consistent with previous literature (Ribet et al. 2010; PMID: 19887568), the expression of the beige markers *PM20D1* and *UCP1* remains significantly lower than in beige adipocytes (**Reviewer Fig. 6**). More importantly, Rosi withdrawal does not impact *PDK4*, *PM20D1* or *UCP1* expression in beige adipocytes (**Reviewer Fig. 6**). These results show that once established, the beige transcriptional state is maintained independently of acute PPAR γ agonism, indicating that beige adipocytes represent a distinct cellular state rather than a transient pharmacological response.

Importantly, however, this does not preclude a role for acutely induced genes in the beige program: for example, *PDK4* is rapidly upregulated by Rosi in white adipocytes, yet in differentiated beige adipocytes *PDK4* knockdown reduces basal oxygen consumption to white cell levels (Barquisseau et al. 2016; PMID: 27110487). Thus, even acutely induced genes can be integral to the functional identity of beige adipocytes, and acute changes in transcription factor binding, chromatin accessibility, and 3D chromatin organization may provide the basis for the long-term establishment and maintenance of the beige state.

Reviewer Fig. 6: RT-qPCR analysis of *PDK4*, *PM20D1* and *UCP1* gene expression in white adipocytes ± 24 h Rosi and beige adipocytes ± Rosi withdrawal

3. In Figure 1, panels C and I, the western blot quantification seems to have been normalized such that the replicate control samples are all set to a value of 1. Since this leads to a non-normal distribution in one of the groups, it seems the authors have chosen to test the statistical significance of the difference between white and beige cells with a Wilcoxon signed ranked test, however it's unclear why all of the data needs to be normalized this way to begin with. Can't the western data for each of the three replicate experiments be normalized to the average densitometry values for the control samples? That would allow the reader to get a sense of the variance inherent to this assay. It's also not entirely clear to me that the data meets the other assumptions of the Wilcoxon test, for example how do the authors see the white and beige conditions as paired data? Maybe because they are cells from the same person, but they've been treated separately so this seems a bit of a stretch. This happens in supplemental 9d too.

We agree and revised the figures 1C and S9D accordingly. Each replicate consists of white and beige parallel differentiation experiments, plated from the same batch of cells, and are thus considered paired samples. A paired t-test was performed after Shapiro-Wilks normality test.

Reviewer Fig. 7: (A) Quantification of FAS, PPARG, CD36, UCP1 and CITED1 protein levels in differentiated (D15) white and beige adipocytes (mean fold difference ± SD; ** $p < 0.01$, ns non-significant, two-tailed paired Student's t test; $n \geq 3$ independent differentiations) (Revised Fig. 1C). (B) Quantification of NFIL3 expression in white and beige adipocytes (D15) (mean fold difference ± SD; ns non-significant, two-tailed paired Student's t test; $n = 3$) (Revised Fig. S9D).

4. The sentence that starts in line 35 is a bit ambiguous, and would make more sense like this (notice the removal of the comma): "We notably uncover NFIL3 as [a] negative regulator of the beige program differentially bound at enhancers of PDK4, a key metabolic switch towards fatty acid oxidation."

The sentence has been edited accordingly.

5. I like the way the heatmap in Figure 2D is labeled on the bottom with the White and Beige and the arrows. Can you label the heatmap in Figure 1F the same way for consistency?

We have relabeled the panels (see Reviewer Fig. 8, revised Fig.1F).

Reviewer Fig. 8: Hierarchical clustering of genes overexpressed in D15 white (wD15 DEGs; left panel) or beige (bD15 DEGs; right panel) adipocytes, scaled across the differentiation time-course (revised Fig.1F)

6. On line 89, the phrase "mitochondrial proteome" struck me a bit. I guess you're setting yourself up for the western blots with the OXPHOS cocktail, but do any of the genes that you're blotting for with this cocktail show differential regulation in the RNA sequencing data, or is the term "Mitochondrial Matrix" showing up in your gene ontology analysis driven by other genes (e.g. transcriptional regulators of OXPHOS genes)? It could be helpful to include the mRNA expression of the genes in the OXPHOS cocktail in a supplemental figure.

To avoid confusion, we have rephrased the sentence as follows:

Line 93: "Interestingly, genes related to distinct mitochondrial processes are overrepresented in both white and beige D15 DEGs (Fig. 1G, Supplemental Fig. S2B), indicating a transcriptional remodeling of the mitochondrial function between beige and white adipocytes in our system."

To directly address the reviewer's question, in addition to the list of differentially expressed genes provided in Table S1, we now include a volcano plot highlighting the expression of genes encoding electron transport chain (ETC) subunits (Reviewer Fig. 9; new Fig. S2B). These data show that several ETC genes are differentially expressed at the mRNA level, indicating that our observations at the protein level are supported by transcriptional regulation.

Reviewer Fig. 9: Volcano plot of differential expression of genes from the mitochondrial respiratory chain in differentiated (D15) white and beige adipocytes (now **Figure S2B**).

7. Line 106-108 was confusing. Is the "...chromatin state model learned from our ChIP seq data..." the thing that you developed with the all of your different ChIP assays (in the methods it says you did ChIP seq for H3K27ac, H3K4me3, H3K4me1, H3K27ac, H3K27me3, H3K36me3, and H3K9me3)? I think it might be helpful to say that these were all done in the results before you say that the DARs map to chromatin state regions defined by these experiments.

We have edited the text to clearly state which histone post-translational modifications were assessed by ChIP-seq:

Line 124: "To assess the epigenetic state of white and beige DARs, we profiled 6 histone post-translational modifications by ChIP-seq: (i) H3K4me3, marking promoters of active genes, (ii) H3K27ac, marking active enhancers and promoters, (iii) H3K4me1, marking enhancers, (iv) H3K36me3, enriched in transcribed gene bodies, (v) H3K27me3, a repressive mark deposited by Polycomb repressive complex 2, and (vi) H3K9me3 marking constitutive heterochromatin."

8. Also in this sentence, if DARs is plural, shouldn't it be "annotate" rather than "annotates"? Line 163 I don't think that regulations should be plural.

Corrected.

9. Like 173 I wasn't clear how the samples were considered biological replicates. It looks like there are two samples labeled "W1" and "W2" and two samples "B1" and "B2" and in the methods they say Hi-ChIP started with 10cm dishes. This sounds like they have technical replicates rather than biological replicates.

For Hi-ChIP, white and beige differentiations were conducted in parallel until day 15, and two independent differentiation experiments were performed. This has been clarified in the methods section:

Line 512: "ASCs were differentiated until D15 in two independent biological replicates."

General information on the number of replicates and how replicates are defined is now included in the Material and methods - *Statistics and reproducibility section*.

10. It would have been excellent to do the C/EBP ChIP PCR similar to what was done with NFIL3. Wonder why this wasn't done?

The footprinting analysis is based on motif accessibility and cannot distinguish which specific C/EBP family member is bound, as several share highly similar motifs. Given that the role of C/EBPs in adipocyte biology is already well established, we focused our efforts on NFIL3, which is less studied in this context. Furthermore, each ChIP experiment requires ~10 million primary cells per condition and replicate, and these cells can only be expanded for a limited number of passages, which imposes significant practical constraints on performing additional transcription factor ChIPs.

Reviewer #4 (Remarks to the Author):

I co-reviewed this manuscript with one of the reviewers who provided the listed reports. This is part of the Communications Biology initiative to facilitate training in peer review and to provide appropriate recognition for Early Career Researchers who co-review manuscripts.

We thank the reviewer for assessing our manuscript.

Reviewer #3 (Remarks to the Author):

1. I concede that IgG isn't a perfect negative control for the NFIL3 chip, but I don't really think the gene desert control is a perfect negative control either. First of all, it's completely unclear where in the genome this primer set is targeted to, and what if these primers don't produce any amplicon at all? I think an absolute minimum for this experiment is to better describe what these control primers are designed to target, preferably with a reference or experimental evidence that they do in fact produce a 116bp amplicon from unprecipitated genomic DNA.

We have now included a genome browser view of the negative control (Ctrl-) primers location (Reviewer Fig. 1A and Supplemental Fig. S10C). These primers are designed within a gene desert, in a region enriched for H3K9me3 and several hundred kb away from a CEBP transcription factor binding site. The primers efficiently amplify the Input of both White and Beige ChIP samples (Reviewer Fig.1B). These primers were originally designed as positive controls for Lamin B1 ChIP-qPCR experiments (Rønningen et al. Genome Res 2015 - Figure 3G amplicon 7); this reference has now been added to the manuscript.

Reviewer Fig. 1. **A** Genome browser views of H3K27ac Hi-ChIP beige loops, ATAC, H3K27ac ChIP, H3K9me3 ChIP, C/EBP transcription factor binding sites identified by ATAC footprinting and primer location of the negative control (Ctrl-) primers. **B** qPCR amplification curve showing detection of the Ctrl- primers amplicon in NFIL3 Inputs and IP samples. The green line indicates the fluorescence threshold used to determine the cycle threshold (Ct) value.

2. Also, if the authors are trying to make the point that there is locus specific binding to enhancer 1 and not enhancer 2 of the PDK gene, shouldn't there be a statistical analysis of these differences? I think it's obvious that there's on average more DNA pulled down from the intronic region and enhancer 1 than there is for enhancer 2, but for the beige cells the binding to enhancer 1 has a relatively large amount of variability, so how do you conclude there is significantly more binding to enh1 than enh2?

We would like to clarify that we do not claim in the manuscript that NFIL3 binding is significantly higher at enhancer 1 than at enhancer 2. The statement refers specifically to the decreased NFIL3 binding at each individual PDK4 regulatory region (intronic and enhancer 1) in beige adipocytes compared to white adipocytes: "NFIL3 binding is significantly decreased at both intronic and distal enhancer of PDK4 in beige adipocytes" (line 271).

As the reviewer notes, Fig. 5G shows that ChIP enrichment at the enhancer 2 region is comparable to that of the negative control, in contrast to the intronic and enhancer 1 regions. Statistical analysis of ChIP results between different PDK4 and control loci confirms that enrichments at both intronic and Enhancer 1 regions are significantly different from the negative control, and from the Enhancer 2 regions (Reviewer Table 1).

	White		Beige	
	Adjusted P Value	Summary	Adjusted P Value	Summary
PDK4_Int. vs. PDK4_Enh.1	0.961	ns	>0.9999	ns
PDK4_Int. vs. PDK4_Enh.2	<0.0001	****	0.0078	**
PDK4_Enh.1 vs. PDK4_Enh.2	<0.0001	****	0.004	**
PDK4_Int. vs. Ctrl-	<0.0001	****	0.0054	**
PDK4_Enh.1 vs. Ctrl-	<0.0001	****	0.0027	**
PDK4_Enh.2 vs. Ctrl-	>0.9999	ns	>0.9999	ns

Reviewer Table 1. Statistical analysis of NFIL3 ChIP-qPCR results. Comparison of ChIP enrichment results (% of input) between different loci (two-way ANOVA with Šídák's multiple comparisons test).

3. Lastly, in the reviewer only figure 6 the authors did the thing where they set all the control samples to 1, so there's no variance in the white samples for any of the 3 genes. That's not possible. I get that it's for reviewers only and they don't specify what test led them to annotate these differences as significant, but I don't understand how they could come to the conclusion there qPCR gives the same value for technical replicates of the same sample.

We have reanalyzed these RT-qPCRs, now expressing all values relative to the average of White samples. Error bars in the revised figure represent the variance among white biological replicates (**Reviewer Fig.2; Supplemental Fig.S2C**). The results and overall conclusion remain unchanged: Rosiglitazone withdrawal does not significantly impact *PDK4*, *PM20D1* or *UCP1* expression in beige adipocytes (Beige vs Beige - Rosi; ns, one-way ANOVA with Tukey's multiple comparisons test).

Reviewer Fig. 2: RT-qPCR analysis of *PDK4*, *PM20D1* and *UCP1* gene expression in white adipocytes \pm 24 h Rosiglitazone 1 μ M and in beige adipocytes \pm 24 h Rosiglitazone withdrawal (mean \pm SD; *** $p < 0.001$, ** $p < 0.01$, * $p < 0.05$, ns non-significant, one-way ANOVA with Tukey's multiple comparisons test; $n = 3$ independent differentiations).